# MeToken: Uniform Micro-environment Token Boosts Post-Translational Modification Prediction

**Cheng Tan**[1,2*], **Zhenxiao Cao**[3*], **Zhangyang Gao**[2*], **Lirong Wu**[2], **Siyuan Li**[2], **Yufei Huang**[2], **Jun Xia**[2], **Bozhen Hu**[2], , **Stan Z. Li**[2†]
[1]Zhejiang University, Hangzhou, China [3]Xi'an Jiaotong University, China
[2]AI Lab, Research Center for Industries of the Future, Westlake University, Hangzhou, China
`{tancheng,gaozhangyang}@westlake.edu.cn; alancao@stu.xjtu.edu.cn`

## Abstract

Post-translational modifications (PTMs) profoundly expand the complexity and functionality of the proteome, regulating protein attributes and interactions that are crucial for biological processes. Accurately predicting PTM sites and their specific types is therefore essential for elucidating protein function and understanding disease mechanisms. Existing computational approaches predominantly focus on protein sequences to predict PTM sites, driven by the recognition of sequence-dependent motifs. However, these approaches often overlook protein structural contexts. In this work, we first compile a large-scale sequence-structure PTM dataset, which serves as the foundation for fair comparison. We introduce the MeToken model, which tokenizes the micro-environment of each amino acid, integrating both sequence and structural information into unified discrete tokens. This model not only captures the typical sequence motifs associated with PTMs but also leverages the spatial arrangements dictated by protein tertiary structures, thus providing a holistic view of the factors influencing PTM sites. Designed to address the long-tail distribution of PTM types, MeToken employs uniform sub-codebooks that ensure even the rarest PTMs are adequately represented and distinguished. We validate the effectiveness and generalizability of MeToken across multiple datasets, demonstrating its superior performance in accurately identifying PTM types. The results underscore the importance of incorporating structural data and highlight MeToken's potential in facilitating accurate and comprehensive PTM predictions, which could significantly impact proteomics research.

## 1 Introduction

Post-translational modifications (PTMs) are chemical modifications that occur to proteins following biosynthesis (Consortium, 2004; Hanahan & Coussens, 2012), playing a pivotal role in regulating biological processes including signal transduction (Meng et al., 2022; Walsh et al., 2005), protein degradation (Jensen, 2004; Craveur et al., 2019; Humphrey et al., 2015), and cellular localization (Li et al., 2021; Deribe et al., 2010). PTMs can alter the properties and functions of a protein by modifying its chemical structure, often serving as switches that toggle protein activity. The diverse nature of PTMs, which include phosphorylation, glycosylation, acetylation, ubiquitination, and methylation, reflects their varied roles across different cellular processes and conditions (Walsh & Jefferis, 2006). Given their complexity and the critical functionality they impart, accurately predicting PTM sites and their specific types is essential for elucidating protein functions, advancing therapeutic development, and unraveling the molecular bases of diseases (Cruz et al., 2019; Tsikas, 2021).

Despite their significance, predicting PTM types remains a formidable challenge due to the inherent complexity of protein structures and the subtle, context-dependent nature of modification sites. Researchers have struggled with several key difficulties: **(i) Predominant computational approaches**

---

*Equal contribution.
†Corresponding author.

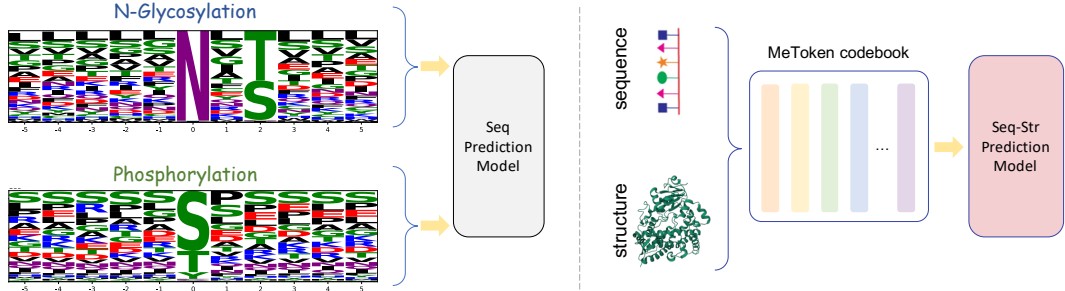

(a) Sequence-based PTM prediction scheme      (b) MeToken PTM prediction scheme

Figure 1: The comparison of sequence-based and MeToken schemes. While sequence-based methods focus on sequence motifs around modification sites, MeToken first encodes the micro-environment at both sequence and structure levels and predicts the PTM types with token embeddings.

**focus on sequence-dependent motifs while neglecting tertiary structural information.** As depicted in Figure 1(a), the N-glycosylation modification exhibits a well-defined 'Asn-X-T/S' sequence motif, a finding thoroughly illustrated by (Bause & Hettkamp, 1979). However, the situation is notably more complex for phosphorylation, where the sequence motifs are less consistent and more variable. The reliance on sequence data alone limits the ability to predict PTM types. **(ii) Lack of large-scale sequence structure PTM dataset.** Though the sequence-structure co-modeling manner has been well explored in protein function prediction (Wang et al., 2022; Hermosilla et al., 2020; Zhang et al., 2022; Fan et al., 2022; Hu et al., 2023), the PTM prediction significantly lacks in terms of a comprehensive, annotated sequence-structure dataset at the residue level. **(iii) The severe long-tail distribution of PTM types.** The distribution of PTM types is highly skewed, with a few types being overwhelmingly more common than others. The lack of representation not only complicates model training with imbalanced data but also hampers the model's generalization ability.

In this paper, we first compile a large-scale sequence-structure PTM dataset featuring over 1.2 million residue-level annotated sites across multiple PTM types. Furthermore, we introduce the MeToken (**M**icro-**e**nvironment Token) model, which integrates both sequence and structural information from the sequence-structure pair. Recognizing that PTMs are influenced not only by the immediate sequence environment but also by the intricate spatial context, MeToken employs a micro-environment-based approach to capture comprehensive contextual information for each amino acid, thereby providing a holistic view that is critical for predicting potential PTMs.

To decipher the typical micro-environment patterns associated with PTMs, we develop a codebook that distills complex and high-dimensional data into simpler, representative tokens with distinct and meaningful representations. The codebook is constructed to learn a set of discrete tokens that correlate specifically to PTMs with minimal redundancy, ensuring that each token captures unique and interpretable information about the micro-environment. Furthermore, we propose a uniform sub-codebook strategy that effectively addresses the long-tail distribution of PTM types, which projects the PTM types into a uniformly distributed token space, ensuring that even the rarest PTMs are adequately represented and distinctly discriminated. By leveraging our tokenization strategy, MeToken not only enhances interpretability but also significantly improves its predictive performance by fine-tuning the model specifically for the PTM prediction task. We conduct extensive experiments to validate the effectiveness and generalizability of MeToken across multiple datasets, demonstrating its superior performance in accurately identifying PTM types.

The contributions of our work are summarized as:

- Recognizing the critical importance of structural context in PTM prediction, we have compiled a large-scale sequence-structure PTM dataset, which serves as the foundational basis for the PTM predictions based on sequence-structure pairs.

- In analogy to sequence motifs used in sequence-based PTM prediction, we introduce the concept of the micro-environment token. The contextual information at both the sequence and structural levels can be uniquely represented within a set of discrete tokens.

- To address the severe long-tail problem prevalent in PTM data, we have devised a codebook construction strategy that incorporates uniform sub-codebooks, ensuring that PTMs are adequately represented and distinctly discernible. Extensive experiments demonstrate the effectiveness and generalizability of MeToken in accurately predicting PTM types.

## 2 RELATED WORK

**Post-Translation Modification Prediction**   Early work on PTM prediction primarily involved manually extracting sequence motifs from around potential modification sites and employing various learning-based methods for prediction (Xue et al., 2010; Lee et al., 2011; Gao et al., 2017; Jia et al., 2016a;b; Luo et al., 2019; Kirchoff & Gomez, 2022; Meng et al., 2022). For example, PPSP (Xue et al., 2006) utilized Bayesian decision theory, focusing on a motif of nine amino acids. LysAcet (Li et al., 2009) used support vector machines (SVM) and sequence coupling patterns. Other methods like PWAA introduced position weight amino acid composition encoding (Qiu et al., 2017), while LightGBM-CroSite combined binary encoding and position weight matrices (Liu et al., 2020). DeepNitro (Xie et al., 2018) incorporated positional amino acid distributions and position-specific scoring matrices. Deep-Kcr integrated sequence-based features and employed convolutional neural networks (CNN) (Lv et al., 2021). DeepSuccinylSite employed one-hot and word embeddings to enhance predictions (Thapa et al., 2020), and Adapt-Kcr combined adaptive embedding with convolutional and long short-term memory networks (LSTM), along with an attention mechanism (Li et al., 2022b). While these methods have shown effectiveness, they are limited by their reliance on sequence motifs, which cannot fully capture the intricate complexity and contextual dependencies of PTM sites. More recent advancements, such as GraphUbiquSite (Chen et al., 2022) and PhosHSGN (Lu et al., 2024), using structural data in predicting Ubiquitylation and Phosphorylation sites. Ertelt et al. (2024) combines PTM prediction with protein design, maximizing or minimizing the predicted probability of a post-translational modification occurring at a specific site. Similarly, other studies focused on PTM crosstalk prediction (Huang & Liu, 2024; Dai et al., 2024), incorporating structural information to address interaction-specific tasks. Despite efforts to integrate sequence and structural data in PTM prediction, the attempt has been largely constrained by the scarcity of comprehensive datasets (Li et al., 2024; Yan et al., 2023a). The largest sequence-structure database, PRISMOID (Li et al., 2020), includes 3,919 protein structure entries with 17,145 non-redundant modification sites. In this work, we compile a large-scale sequence-structure PTM dataset with over 1.2 million annotated sites across over 180,000 proteins.

**Sequence-structure Co-modeling**   Sequence-structure co-modeling has become a common manner in predicting protein property and function. LM-GVP (Wang et al., 2021) combined protein language model and graph neural networks (GNN) to predict protein properties from both their amino acid sequences and 3D structures. DeepFRI (Gligorijević et al., 2021) employed a similar strategy and used homology models. GearNet (Zhang et al., 2022) is a seminal work that pretraining protein representations based on 3D structures using a graph encoder and multiview contrastive learning. ProNet (Wang et al., 2022) employed a hierarchical graph network to learn detailed and granular protein representations. CDConv (Fan et al., 2022) utilized independent weights for regular sequential displacements and direct encoding for irregular geometric displacements. While these methods utilized local sequence-structure patterns to learn global protein-level properties, PTM prediction differs fundamentally as it directly employs the local sequence-structure micro-environments to predict modifications at the residue level.

**Micro-environment Tokenization**   Vector quantization (VQ) is a pivotal technique for efficient data compression and representation. The basic premise involves compressing latent representations by mapping them to the nearest vector within a codebook. Vanilla VQ (Van Den Oord et al., 2017) straightforwardly assigns each vector in the latent space to the closest codebook vector. Product Quantization (Jégou et al., 2011; Chen et al., 2020; El-Nouby et al., 2022) segments the high-dimensional space into a Cartesian product of lower-dimensional subspaces. Residual quantization (Juang & Gray, 1982; Martinez et al., 2014; Lee et al., 2022; Zeghidour et al., 2021) iteratively quantizes vectors and their residuals, representing the vector as a stack of codes. Lookup-free VQ (Mentzer et al., 2023; Yu et al., 2023b) quantizes each dimension to a fixed set of values. Although advancements in vector quantization have shown promise in molecular modeling (Xia et al., 2022; Gao et al., 2024; Wu et al., 2024), the specific application of micro-environment tokenization in PTM prediction remains underexplored, particularly given the challenges posed by the long-tail distribution of PTMs.

## 3 PRELIMINARIES

A protein $\mathcal{P}$ is composed of $N$ amino acids, which are organized as a string of primary sequence, denoted as $\mathcal{S} = \{s_i\}_{i=1}^{N}$. Each element $s_i$ within this sequence is referred to as a residue, representing one of the 20 standard amino acid types. In the real physicochemical environment of a cell, these

sequences fold into stable three-dimensional structures. This three-dimensional conformation is defined by the coordinates of backbone atoms, represented as $\mathcal{X} = \{\boldsymbol{x}_{i,\omega}\}_{i=1}^{N}$, where $\boldsymbol{x}_{i,\omega} \in \mathbb{R}^3$ and $\omega$ encompasses the main backbone atoms, specifically $\{C\alpha, N, C, O\}$. The sequence-structure pair of a protein is represented as $\mathcal{P} = (\mathcal{S}, \mathcal{X})$.

Modeling the sequence and structure of a protein $\mathcal{P}$ as a residue-level protein graph $\mathcal{G} = (\mathcal{V}, \mathcal{E})$ is a widely recognized approach (Tan et al., 2024; Luo et al., 2022; Kong et al., 2023a; Jin et al., 2020; Kong et al., 2023b). In this representation, $\mathcal{V}$ represents the node embeddings, which are derived from the amino acid sequence $s_i$ and the spatial arrangement $\{\boldsymbol{x}_{i,\omega} | \omega \in \{C\alpha, N, C, O\}\}$ of the backbone atoms within each residue. The edges, denoted as $\mathcal{E}$, encapsulate the interactions between residues, represented as edge embeddings that reflect the various types of connections. The node and edge embedding construction is detailed in the Appendix B. Building on the concept of the micro-environment, as outlined in previous works (Huang et al., 2023; Lu et al., 2022; Wu et al., 2024), the micro-environment of a residue $i$ within $\mathcal{G}$ is characterized by its node set $V_{\text{ME}}^i \subseteq \mathcal{V}$, can be formally defined as:

$$V_{\text{ME}}^i = \left\{ v_j \mid |i - j| \leq d_s, \|C\alpha_i - C\alpha_j\| \leq d_r, v_j \in \mathcal{N}_i^{(K)}, \forall j \right\}, \tag{1}$$

where $d_s$ and $d_r$ represent the cut-off distances within the sequence and in 3D space, respectively. $C\alpha_i$ and $C\alpha_j$ denote the 3D coordinates of the carbon-alpha atoms in residues $i$ and $j$, and $\mathcal{N}_i^{(K)}$ indicates the $K$-hop neighborhood of residue $i$ in the 3D structure.

Thus, the types of edges in the micro-environment focus on three specific interactions: *(i) sequential edge*, connecting two residues in the sequence within a cutoff distance $d_s$; *(ii) radius edge*, connecting two residues if the spatial Euclidean distance between them is less than a threshold $d_r$; and *(iii) K-nearest edge*, connecting two residues that are among the spatial $K$-nearest neighbors to each other. Consequently, the complete micro-environment of residue $i$ is represented as $\mathcal{G}_{\text{ME}}^i = (V^i, E_{\text{ME}}^i)$, where $V_i \in \mathcal{V}$ includes the node embedding of the residue $i$ and $E_{\text{ME}}^i$ includes the edge embeddings connecting residue $i$ with its neighboring nodes from three kinds of edges in the set $V_{\text{ME}}^i$.

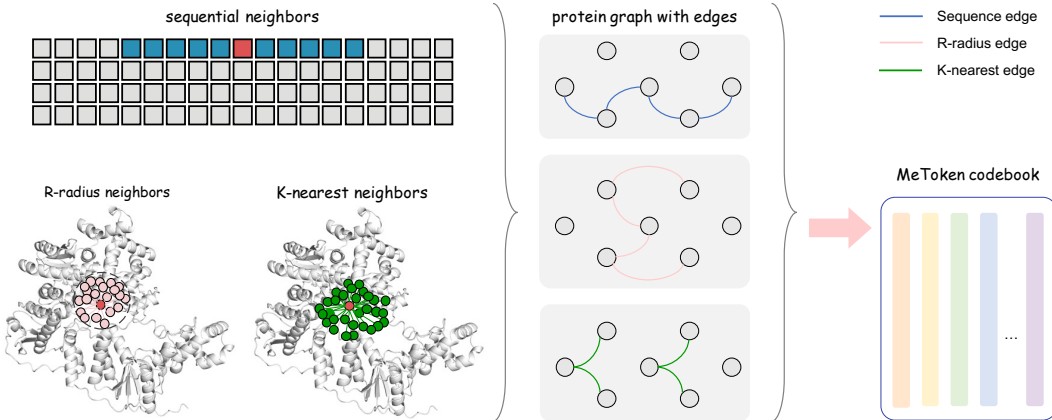

Figure 2: Illustration of the micro-environment of residue $i$. This figure depicts the local neighborhood of residue $i$, highlighted in red, which is interconnected through various types of edges: sequential edges (blue), $R$-radius edges (pink), and $K$-nearest edges (green). The entire micro-environment, including residue $i$ and its interconnected neighbors, is then tokenized into a unified representation token for PTM prediction.

In this study, our primary objective is to effectively encode the micro-environment of each residue $i$ within the protein graph $\mathcal{G}$, by seamlessly integrating crucial sequence and structural information into a single, unified discrete token. As depicted in Figure 2, the micro-environment of residue $i$ is adeptly tokenized into a unified representation token. The core of our tokenization process involves developing a mapping function $\mathcal{F} : \mathcal{G}_{\text{ME}}^i \rightarrow \boldsymbol{e}^j$, where $\boldsymbol{e}^j \in \mathbb{R}^d$ denotes a $d$-dimensional token embedding. This embedding is not merely a numerical representation but a rich encapsulation of the complex patterns of sequence and spatial features that characterize the unique micro-environment surrounding residue $i$. Here, $j$ signifies the token index within the codebook $\mathcal{C}$, ranging over $[1, |\mathcal{C}|] \cap \mathbb{Z}$.

## 4 METOKEN

In this section, we detail the methodologies underlying our proposed MeToken model, which aims to tackle the challenges posed by the diverse nature and varying frequency of PTM types observed across different proteins. The MeToken model is specifically designed to manage the long-tail distribution of PTM types and capture subtle, yet crucial, biochemical signals in protein micro-environments.

### 4.1 UNIFORM SUB-CODEBOOK CONSTRUCTION

Recognizing that the distribution of PTM types exhibits a severe long-tail pattern, where many PTM types are infrequently observed, it is imperative to ensure that rare PTMs are adequately represented in the learning process. To achieve this, we allocate an identical size of sub-codebook to each PTM type, irrespective of their frequency in the dataset. This uniformity ensures that no PTM type is inherently disadvantaged by a lack of representative power within the model's architecture.

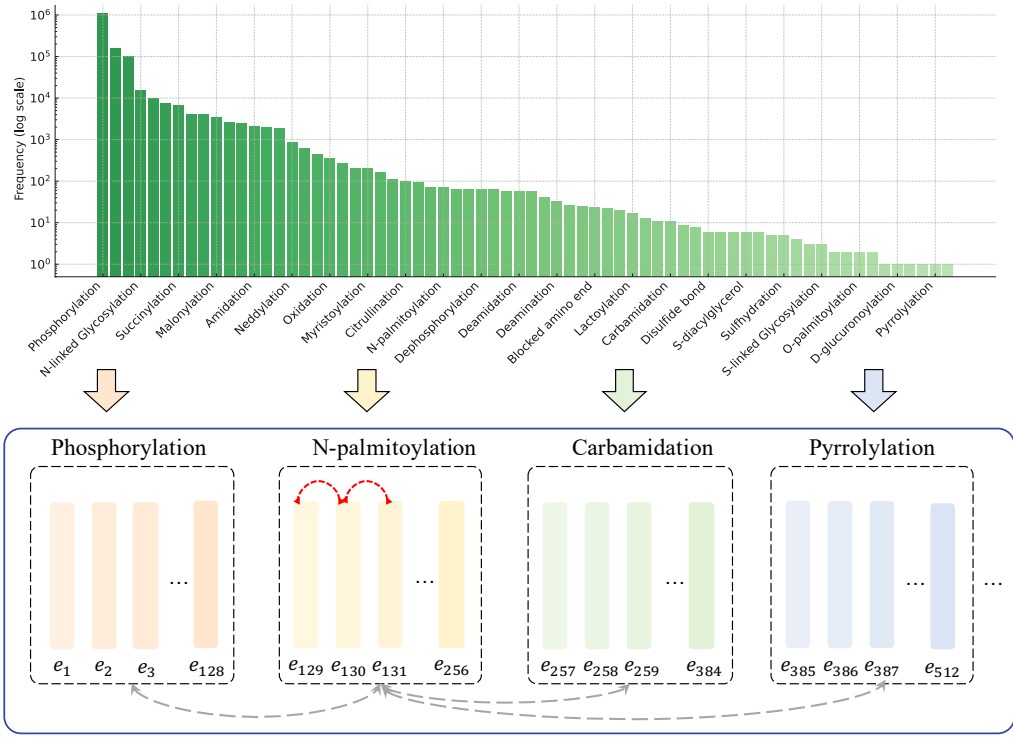

Figure 3: The long-tail distributed PTM types are projected into a uniformly distributed token space. Red arrows depict the consolidation of token embeddings within each sub-codebook, enhancing intra-class similarity. In contrast, gray arrows represent the dispersion of token embeddings across different sub-codebooks, promoting inter-class distinctiveness.

As illustrated in Figure 3, each sub-codebook is designed to capture distinct patterns that are characteristic of specific PTM types. However, merely maintaining a uniform size for each sub-codebook does not inherently prevent the potential risk of token embeddings collapsing into non-discriminative clusters. To address this, we implement a uniform loss mechanism specifically designed to refine the distribution of token embeddings, which is formally defined as follows:

$$\mathcal{L}_u = -\frac{1}{|\mathcal{C}|} \sum_{i=1}^{|\mathcal{C}|} \log \frac{\sum_{j=1} C_{ij} \mathbb{1}_{S_i=S_j}}{\sum_{j=1} C_{ij}}, \tag{2}$$

where $S_i$ denotes the PTM type of token $i$, $C_{ij} = \exp(\frac{e_i^T e_j}{\|e_i\|\|e_j\|}/\tau_u)$ represents the cosine similarity between tokens $i$ and $j$ scaled by the temperature $\tau_u$, and $\mathbb{1}_{S_i=S_j}$ is an indicator that is 1 if token $i$ and $j$ share the same PTM type, and 0 otherwise. This formulation maximizes intra-class similarities, thereby enhancing the homogeneity of embeddings within each PTM class. Simultaneously, it minimizes inter-class similarities, promoting heterogeneity across different PTM types.

## 4.2 TEMPERATURE-SCALED VECTOR QUANTIZATION

Vanilla vector quantization (Van Den Oord et al., 2017) maps each input vector to the nearest vector in a predefined set, known as a codebook. This mapping results in a "hard assignment" that can be represented as:

$$z_i = \arg\min_j \|\boldsymbol{h}_i - \boldsymbol{e}_j\|^2, \tag{3}$$

where $\boldsymbol{h}_i \in \mathbb{R}^d$ represents the input vector and $\boldsymbol{e}_j$ denotes the nearest codebook vector. The index $z_i$ identifies the specific codebook vector that minimizes the Euclidean distance to $\boldsymbol{h}_i$. A fundamental limitation of this method is the non-differentiable nature of the mapping process due to the hard assignment, which relies on the nearest neighbor search. This search uses the encoder output to find the closest code, thereby rendering the quantization step non-differentiable:

$$\mathcal{L}_{vq} = \mathcal{L}_{recon} + \|\mathrm{sg}[\boldsymbol{h}_i] - \boldsymbol{e}_{z_i}\|_2^2 + \beta\|\boldsymbol{h}_i - \mathrm{sg}[\boldsymbol{e}_{z_i}]\|_2^2, \tag{4}$$

where $\mathcal{L}_{recon}$ is the reconstruction loss. $\mathrm{sg}[\cdot]$ indicates the stop-gradient operation, and $\beta$ is the trade-off hyperparameter. However, the rigid and non-differentiable characteristics of hard quantization may not effectively capture the subtle and complex variations typical in PTM datasets, particularly in the context of our model which employs uniform sub-codebook construction. These challenges can lead to gradient sparsity and impede the learning process, necessitating a more flexible approach.

In our model, we implement a temperature-scaled vector quantization mechanism that introduces a temperature parameter, $\tau_v$, to modulate the quantization process. The quantization process can be mathematically defined as follows:

$$\hat{\boldsymbol{h}}_i = \sum_{j=1}^{|\mathcal{C}|} a_{ij}\boldsymbol{e}_j, \quad a_{ij} = \frac{\exp(\boldsymbol{h}_i^T \boldsymbol{e}_j / \tau_v)}{\sum_{k=1}^{|\mathcal{C}|} \exp(\boldsymbol{h}_i^T \boldsymbol{e}_k / \tau_v)}, \tag{5}$$

where $a_{ij}$ represents the attention weight assigned to the $j$-th codebook vector for the $i$-th input vector. These weights are computed using a softmax function, scaled by the temperature parameter $\tau_v$, which modulates the sharpness of the resulting attention distribution. A higher $\tau_v$ leads to a more diffuse distribution across the codebook vectors, promoting a broader exploration of the vector space and mitigating the risk of premature convergence to suboptimal quantizations. Initially set at 1, $\tau_v$ is gradually reduced towards zero during training. As $\tau_v$ diminishes, $a_{ij}$ transitions towards a one-hot distribution, rendering the quantization process increasingly deterministic. The quantization selects the codebook vector corresponding to the highest attention weight, identified by $z_i = \arg\max_j a_{ij}$. Tthe codebook training loss incorporates both the reconstruction loss and a uniform loss:

$$\mathcal{L}_{codebook} = \mathcal{L}_{recon} + \alpha\mathcal{L}_u \tag{6}$$

where $\alpha$ is set as 0.1 empirically, balancing the reconstruction loss and the uniform loss.

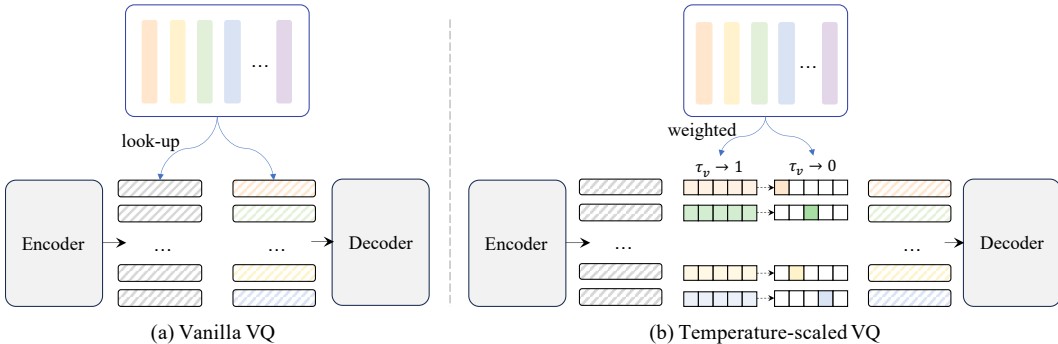

(a) Vanilla VQ      (b) Temperature-scaled VQ

Figure 4: Vanilla VQ looks up the codebook and hard-assigns the nearest code, while temperature-scaled VQ employs a softer, probabilistic assignment approach where codebook vectors are assigned based on weights. These weights are modulated by $\tau_v$, which adjusts the sharpness throughout the training process, transitioning from exploratory to more deterministic assignments as $\tau_v$ decreases.

### 4.3 PTM Prediction with Learned Codebook

The learned codebook, derived from the training set, contains vectors that capture the unique attributes of protein micro-environments. As depicted in Figure 5, during the prediction phase, we extract the codebook vector $\boldsymbol{e}_{z_i}$ corresponding to the quantized index $z_i$ associated with each residue's micro-environment. This retrieval initiates the prediction process. To predict the specific PTM type for each residue, we employ a prediction network built from three layers of PiGNN (Gao et al., 2022), a graph neural network tailored for protein graph data. This network directly processes the quantized codebook vectors $\boldsymbol{e}_{z_i}$ transforming these richly encoded representations into predictions about PTM types. The predictor network is trained using the cross-entropy loss, which is defined as:

$$\mathcal{L}_{pred} = -\sum_i \sum_c y_{ic} \log p(y_{ic}|\boldsymbol{h}_i), \tag{7}$$

where $y_{ic}$ indicates whether the $i$-th site has the PTM type $c$, $\boldsymbol{h}_i$ is the micro-environment embedding.

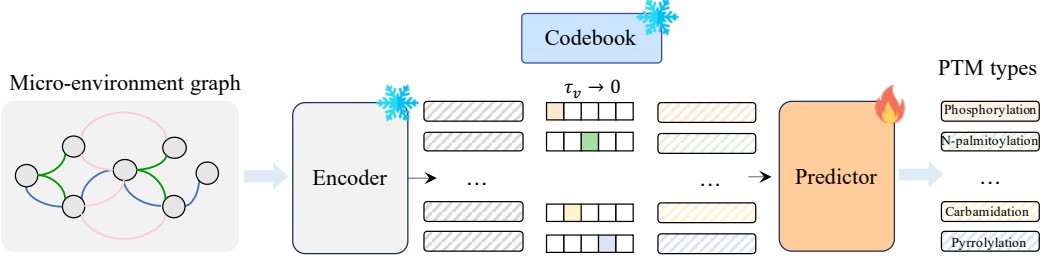

Figure 5: The PTM prediction process with the learned codebook.

## 5 Experiment

**Datasets** Experiments are conducted on three datasets: *our large-scale dataset*, *PTMint*, and *qPTM*. We constructed a large-scale dataset by integrating dbPTM (Li et al., 2022a), the most extensive sequence-based PTM dataset available, with structural data obtained from the Protein Data Bank (PDB) (Berman et al., 2000) and the AlphaFold database (Varadi et al., 2022; 2024). We utilized MMseqs2 (Steinegger & Söding, 2017) to cluster the data based on sequence similarity with a threshold of 40% and grouped the data into clusters, which were then allocated to the training, validation, or test set. Our large-scale PTM dataset with over 1.2 million annotated sites across over 180,000 proteins is the largest PTM dataset with both sequence and structure data. To assess the generalizability, we used the pre-trained models on the large-scale dataset to directly test on the PTMint (Hong et al., 2023) and qPTM (Yu et al., 2023a) datasets. The details of the dataset construction and the statistics of open-sourced datasets are outlined in Appendix C.

**Baselines** We compare the performance against a selection of established PTM prediction models. Specifically, we include sequence-based baselines such as EMBER(Kirchoff & Gomez, 2022), DeepPhos(Luo et al., 2019), and MusiteDeep (Wang et al., 2017). MIND (Yan et al., 2023a;b) is as a sequence-structure baseline. *In addition to these PTM-specific models, we incorporate baselines from broader protein modeling domains.* For structure-based baselines, we adapt StructGNN (Ingraham et al., 2019), GraphTrans (Ingraham et al., 2019), GVP (Jing et al., 2020), and PiFold (Gao et al., 2022), which are primarily designed for structure-based protein sequence design. Similarly, ESM-2(Lin et al., 2023) is fine-tuned to serve as an additional sequence-based PTM predictor. Moreover, GearNet (Zhang et al., 2022) and CDConv (Fan et al., 2022), originally designed for protein function prediction, are also adapted to act as sequence-structure co-modeling baselines.

**Metrics** Regarding the PTM prediction problem as a multi-class classification problem, we employed a comprehensive suite of metrics, such as accuracy, precision, recall, F1-score, MCC, AUROC, and AUPRC within our multi-class context. We detail these metrics in the Appendix D.

### 5.1 MeToken Outperforms Baseline Models on The Large-scale Dataset

To evaluate the performance of MeToken, we conducted extensive tests on our large-scale dataset that incorporates both sequence and structural data. Table 1 presents the performance comparison of MeToken against a range of baseline models. It is noteworthy that both sequence-based and structure-based baselines exhibit comparable performances, each utilizing only a single modality.

Among these, the finetuned ESM-2 emerged as the most robust, achieving the best performance across all baselines. Despite its strong performance, MeToken demonstrates superior capabilities, achieving a remarkable 6.46% improvement in F1-score over ESM-2. This significant performance underscores the efficacy of MeToken in leveraging micro-environment information from sequence-structure data.

Table 1: Performance comparison on the large-scale dataset. The mean/std of three runs is reported.

| Input | Method | F1-score | Accuracy | Precision | Recall | MCC | AUROC | AUPRC |
|---|---|---|---|---|---|---|---|---|
| Seq | EMBER | $10.94_{\pm 0.77}$ | $72.89_{\pm 0.12}$ | $15.34_{\pm 0.94}$ | $9.55_{\pm 0.67}$ | $19.75_{\pm 1.07}$ | $64.47_{\pm 2.12}$ | $9.89_{\pm 0.71}$ |
| | DeepPhos | $3.58_{\pm 0.00}$ | $74.33_{\pm 0.01}$ | $5.75_{\pm 0.18}$ | $4.18_{\pm 0.01}$ | $2.96_{\pm 0.09}$ | $59.34_{\pm 1.42}$ | $5.17_{\pm 0.16}$ |
| | MusiteDeep | $3.55_{\pm 0.00}$ | $74.18_{\pm 0.06}$ | $3.09_{\pm 0.00}$ | $4.17_{\pm 0.00}$ | $0.00_{\pm 0.00}$ | $48.85_{\pm 1.04}$ | $4.01_{\pm 0.01}$ |
| | ESM-2 | $43.94_{\pm 0.07}$ | $89.75_{\pm 0.01}$ | $55.67_{\pm 1.94}$ | $42.15_{\pm 0.04}$ | $75.83_{\pm 0.01}$ | $91.52_{\pm 0.03}$ | $47.31_{\pm 0.14}$ |
| Str | StructGNN | $12.86_{\pm 0.06}$ | $75.62_{\pm 0.08}$ | $21.16_{\pm 0.61}$ | $11.06_{\pm 0.18}$ | $29.73_{\pm 0.26}$ | $78.94_{\pm 0.46}$ | $13.51_{\pm 0.18}$ |
| | GraphTrans | $5.03_{\pm 0.49}$ | $74.14_{\pm 0.11}$ | $12.23_{\pm 0.72}$ | $5.12_{\pm 0.40}$ | $9.87_{\pm 1.82}$ | $74.75_{\pm 0.08}$ | $7.99_{\pm 0.27}$ |
| | GVP | $17.51_{\pm 1.12}$ | $77.62_{\pm 0.16}$ | $32.19_{\pm 4.87}$ | $14.43_{\pm 0.92}$ | $37.95_{\pm 0.89}$ | $82.47_{\pm 0.52}$ | $18.83_{\pm 1.29}$ |
| | PiFold | $8.87_{\pm 4.69}$ | $75.11_{\pm 1.37}$ | $18.42_{\pm 5.43}$ | $8.03_{\pm 3.71}$ | $22.59_{\pm 10.15}$ | $76.81_{\pm 4.37}$ | $11.02_{\pm 4.43}$ |
| Seq + Str | GearNet | $16.63_{\pm 1.64}$ | $85.75_{\pm 1.13}$ | $17.09_{\pm 1.37}$ | $21.01_{\pm 1.45}$ | $67.64_{\pm 2.44}$ | $89.37_{\pm 0.29}$ | $21.79_{\pm 0.25}$ |
| | CDConv | $18.11_{\pm 0.62}$ | $87.29_{\pm 0.09}$ | $19.18_{\pm 1.97}$ | $22.58_{\pm 1.52}$ | $70.88_{\pm 0.16}$ | $89.42_{\pm 0.21}$ | $21.94_{\pm 0.41}$ |
| | MIND | $26.70_{\pm 0.72}$ | $86.30_{\pm 0.24}$ | $33.35_{\pm 4.07}$ | $25.21_{\pm 0.84}$ | $66.63_{\pm 0.48}$ | $77.37_{\pm 1.74}$ | $22.29_{\pm 0.36}$ |
| | MeToken | $\mathbf{50.40}_{\pm 0.06}$ | $\mathbf{89.91}_{\pm 0.05}$ | $\mathbf{58.08}_{\pm 0.38}$ | $\mathbf{47.93}_{\pm 0.14}$ | $\mathbf{76.03}_{\pm 0.06}$ | $\mathbf{92.20}_{\pm 0.03}$ | $\mathbf{51.26}_{\pm 0.14}$ |

## 5.2 METOKEN GENERALIZE WELL ON PTMINT AND QPTM

To further evaluate the effectiveness and generalizability of MeToken, we conducted rigorous tests on two additional datasets, PTMint and qPTM, after training the models on our large-scale dataset. We refined the qPTM dataset by excluding data with sequence similarity greater than 40% compared to our training set to prevent data leakage, while the PTMint dataset remained unfiltered due to its relatively small size. The summarized results in Tables 2 and 3 highlight MeToken's consistent superiority over all baselines across various metrics on both datasets.

The PTMint dataset, although smaller in scale, contains a well-curated collection of samples spread across several protein types, making it an ideal testbed for evaluating PTM predictions. The results from Table 2 demonstrate that MeToken significantly outperforms sequence-only and structure-only models, showcasing its capability to integrate micro-environments from complex data modalities.

Table 2: Performance comparison on the PTMint dataset. The mean/std of three runs is reported.

| Input | Method | F1-score | Accuracy | Precision | Recall | MCC | AUROC | AUPRC |
|---|---|---|---|---|---|---|---|---|
| Seq | EMBER | $21.27_{\pm 3.81}$ | $81.53_{\pm 1.05}$ | $21.98_{\pm 4.91}$ | $21.40_{\pm 2.34}$ | $8.16_{\pm 1.11}$ | $66.60_{\pm 2.24}$ | $30.23_{\pm 1.21}$ |
| | DeepPhos | $19.32_{\pm 1.20}$ | $85.97_{\pm 1.50}$ | $18.55_{\pm 0.85}$ | $20.68_{\pm 2.34}$ | $-2.00_{\pm 2.99}$ | $53.64_{\pm 2.06}$ | $22.66_{\pm 0.79}$ |
| | MusiteDeep | $19.01_{\pm 0.01}$ | $90.54_{\pm 0.11}$ | $18.11_{\pm 0.02}$ | $20.00_{\pm 0.00}$ | $0.00_{\pm 0.00}$ | $49.70_{\pm 2.66}$ | $22.11_{\pm 1.01}$ |
| | ESM-2 | $36.46_{\pm 2.23}$ | $93.07_{\pm 0.59}$ | $\mathbf{41.78}_{\pm 7.87}$ | $36.30_{\pm 1.05}$ | $61.50_{\pm 3.41}$ | $97.79_{\pm 0.09}$ | $60.82_{\pm 2.02}$ |
| Str | StructGNN | $20.55_{\pm 1.66}$ | $87.29_{\pm 0.78}$ | $21.69_{\pm 1.45}$ | $20.23_{\pm 1.71}$ | $13.52_{\pm 4.98}$ | $77.74_{\pm 0.04}$ | $33.71_{\pm 2.60}$ |
| | GraphTrans | $20.55_{\pm 3.43}$ | $87.56_{\pm 1.45}$ | $19.53_{\pm 3.15}$ | $22.18_{\pm 4.34}$ | $5.91_{\pm 5.47}$ | $63.50_{\pm 2.30}$ | $26.41_{\pm 1.90}$ |
| | GVP | $25.00_{\pm 4.64}$ | $87.41_{\pm 1.49}$ | $25.23_{\pm 4.17}$ | $26.11_{\pm 5.43}$ | $29.56_{\pm 6.04}$ | $76.88_{\pm 1.59}$ | $33.17_{\pm 0.76}$ |
| | PiFold | $20.32_{\pm 3.28}$ | $89.19_{\pm 0.15}$ | $21.54_{\pm 4.76}$ | $20.59_{\pm 3.05}$ | $13.28_{\pm 4.21}$ | $71.02_{\pm 0.89}$ | $27.73_{\pm 0.83}$ |
| Seq + Str | GearNet | $32.09_{\pm 0.24}$ | $94.31_{\pm 0.16}$ | $28.66_{\pm 0.25}$ | $40.00_{\pm 0.00}$ | $68.23_{\pm 0.55}$ | $97.69_{\pm 0.12}$ | $56.47_{\pm 1.32}$ |
| | CDConv | $29.30_{\pm 3.77}$ | $92.75_{\pm 0.65}$ | $26.62_{\pm 3.59}$ | $39.50_{\pm 2.68}$ | $59.98_{\pm 2.17}$ | $97.83_{\pm 0.29}$ | $56.97_{\pm 0.96}$ |
| | MIND | $35.04_{\pm 1.49}$ | $93.25_{\pm 0.58}$ | $34.07_{\pm 4.53}$ | $38.46_{\pm 1.32}$ | $64.40_{\pm 2.25}$ | $81.30_{\pm 3.50}$ | $51.56_{\pm 1.19}$ |
| | MeToken | $\mathbf{42.07}_{\pm 1.26}$ | $\mathbf{94.51}_{\pm 0.15}$ | $38.74_{\pm 0.60}$ | $\mathbf{49.59}_{\pm 1.41}$ | $\mathbf{70.92}_{\pm 1.45}$ | $\mathbf{98.55}_{\pm 0.19}$ | $\mathbf{73.22}_{\pm 2.57}$ |

On the other hand, the qPTM dataset provides a more expansive evaluation landscape due to its large volume of samples and diversity of protein types, making it perfectly suited for assessing the scalability and adaptability of PTM models. As illustrated in Table 3, MeToken not only excels in performance but also consistently achieves superior metrics across all evaluated categories when compared to competing models. The strong performance of MeToken across these varied datasets underscores its capacity to generalize well beyond the confines of its initial training environment. This trait is especially crucial in proteomics, where predictive models must reliably perform across a wide array of unseen data, encompassing diverse biological contexts and experimental conditions.

Table 3: Performance comparison on the qPTM dataset. The mean/std of three runs is reported.

| Input | Method | F1-score | Accuracy | Precision | Recall | MCC | AUROC | AUPRC |
|---|---|---|---|---|---|---|---|---|
| Seq | EMBER | $21.07_{\pm1.60}$ | $67.86_{\pm1.53}$ | $28.37_{\pm2.43}$ | $21.91_{\pm1.61}$ | $17.41_{\pm2.95}$ | $58.89_{\pm2.24}$ | $28.88_{\pm0.57}$ |
| | DeepPhos | $17.85_{\pm1.13}$ | $65.36_{\pm1.39}$ | $18.08_{\pm2.49}$ | $20.33_{\pm0.79}$ | $1.42_{\pm3.91}$ | $47.52_{\pm2.68}$ | $20.98_{\pm1.12}$ |
| | MusiteDeep | $16.04_{\pm0.09}$ | $66.98_{\pm0.63}$ | $13.39_{\pm0.12}$ | $20.00_{\pm0.00}$ | $0.00_{\pm0.00}$ | $46.42_{\pm4.46}$ | $20.81_{\pm0.95}$ |
| | ESM-2 | $57.40_{\pm1.38}$ | $86.79_{\pm1.15}$ | $61.57_{\pm2.13}$ | $56.47_{\pm0.91}$ | $74.26_{\pm2.22}$ | $94.91_{\pm1.02}$ | $60.80_{\pm2.63}$ |
| Str | StructGNN | $23.90_{\pm3.27}$ | $71.45_{\pm1.44}$ | $36.40_{\pm4.25}$ | $23.87_{\pm2.89}$ | $31.60_{\pm2.58}$ | $82.25_{\pm0.35}$ | $46.12_{\pm2.54}$ |
| | GraphTrans | $20.85_{\pm3.49}$ | $68.95_{\pm1.59}$ | $29.21_{\pm8.70}$ | $22.07_{\pm2.76}$ | $21.20_{\pm3.88}$ | $75.77_{\pm4.02}$ | $36.54_{\pm0.91}$ |
| | GVP | $28.31_{\pm2.32}$ | $73.24_{\pm0.62}$ | $40.66_{\pm7.84}$ | $26.76_{\pm2.02}$ | $38.95_{\pm2.08}$ | $83.56_{\pm1.69}$ | $50.94_{\pm2.95}$ |
| | PiFold | $22.29_{\pm3.10}$ | $69.37_{\pm1.27}$ | $31.53_{\pm12.39}$ | $23.73_{\pm2.00}$ | $22.99_{\pm7.39}$ | $70.69_{\pm1.53}$ | $35.35_{\pm1.15}$ |
| Seq + Str | GearNet | $31.05_{\pm0.26}$ | $78.95_{\pm0.75}$ | $27.68_{\pm0.25}$ | $39.90_{\pm0.00}$ | $61.16_{\pm0.82}$ | $92.31_{\pm0.90}$ | $54.49_{\pm0.78}$ |
| | CDConv | $45.20_{\pm3.20}$ | $81.12_{\pm1.35}$ | $47.68_{\pm0.25}$ | $51.23_{\pm3.31}$ | $66.25_{\pm2.24}$ | $92.99_{\pm0.27}$ | $56.65_{\pm0.97}$ |
| | MIND | $43.88_{\pm11.54}$ | $84.46_{\pm0.54}$ | $50.10_{\pm4.11}$ | $43.01_{\pm10.99}$ | $67.65_{\pm1.68}$ | $84.13_{\pm2.31}$ | $51.34_{\pm1.19}$ |
| | **MeToken** | $\mathbf{61.95}_{\pm1.07}$ | $\mathbf{88.39}_{\pm1.04}$ | $\mathbf{64.97}_{\pm1.14}$ | $\mathbf{60.40}_{\pm1.17}$ | $\mathbf{77.56}_{\pm1.44}$ | $\mathbf{95.88}_{\pm0.52}$ | $\mathbf{64.05}_{\pm1.19}$ |

## 5.3 ABLATION STUDY

To highlight the impact of components, we compare MeToken against several baseline configurations. The results, as detailed in Table 4, offer a depiction of how each element contributes to the overall effectiveness: (i) Micro-env (Me): This baseline employs an end-to-end model that considers micro-environments without tokenization. Serving as a fundamental benchmark, it attains an F1-score of 40.04%. (ii) Me + Uni-Codebook: By integrating micro-environment tokenization with a uniform sub-codebook that acknowledges the long-tail distribution of PTM types, this configuration yields a substantial enhancement, elevating the F1-score to 48.04%. (iii) Me + TS-VQ: This setup explores the use of an unconstrained codebook paired with temperature-scaled vector quantization (TS-VQ). Despite TS-VQ's theoretical benefits—softer assignments and broader exploration—this configuration disappointingly results in an F1-score of only 37.46%.

These findings underscore the crucial necessity for a well-structured codebook; in the absence of such a framework, the inherent advantages of TS-VQ cannot be fully realized, particularly when navigating the complexities introduced by the long-tail distribution of PTM types. Ultimately, our comprehensive implementation of MeToken, which synergistically combines micro-environment tokenization with a uniform sub-codebook and TS-VQ, achieves the highest performance, attaining an impressive F1-score of 50.40%. This result illustrates the effectiveness of our integrated approach in maximizing the model's potential.

Table 4: The ablation of MeToken on the large-scale dataset.

| Method | Accuracy | Precision | Recall | F1-score | MCC | AUROC | AUPRC |
|---|---|---|---|---|---|---|---|
| Micro-env (Me) | 88.82 | 49.09 | 37.81 | 40.04 | 73.48 | 91.98 | 42.95 |
| Me + Uni-Codebook | 88.74 | 51.95 | 47.70 | 48.04 | 73.37 | 92.32 | 48.62 |
| Me + TS-VQ | 88.22 | 41.29 | 38.62 | 37.46 | 72.54 | 91.25 | 40.51 |
| MeToken | 89.91 | 58.08 | 47.93 | 50.40 | 76.03 | 92.20 | 51.26 |

## 5.4 VISUALIZATION AND BIOLOGICAL UNDERSTANDING OF OUR MODEL

We visualize the code embeddings learned by MeToken using t-SNE (Van der Maaten & Hinton, 2008) in Figure 6. Due to space constraints, we used abbreviated labels, with their full names available in Appendix G. It reveals three key observations that offer biologically relevant insights:

① **Relationship between different types of Glycosylation:** C-glycosylation is positioned notably far from N- and O-glycosylation. Unlike N- and O-glycosylation, where the sugar attaches to nitrogen (N) or oxygen (O) atoms, C-glycosylation involves the formation of a covalent bond between the sugar and a carbon atom (Ihara et al., 2015; Lovelace et al., 2011). This spatial separation in the embedding space aligns with the fundamental differences in their underlying biochemical mechanisms.

② **Competitive modification between phosphorylation and acetylation sites:** The scattered distribution of phosphorylation and acetylation sites in the plot reflects the competitive nature of these PTMs (Shukri et al., 2023; Csizmok & Forman-Kay, 2018). Both modifications often occur within the same microenvironment and compete for access to nearby residues, resulting in the observed pattern where these two PTMs are dispersed, sometimes adjacent or interleaved.

③ **Distribution of phosphorylation and ubiquitination sites** These sites are distributed in a dispersed manner, lacking a central cluster but instead spread across several distinct regions. This fragmented pattern aligns with observations from previous studies (Shi, 2009)(Johnson & Lewis, 2001)(Johnson, 2009), which emphasize the intricate interplay between these two PTMs. They often function in a coordinated fashion, regulating diverse aspects of protein behavior.

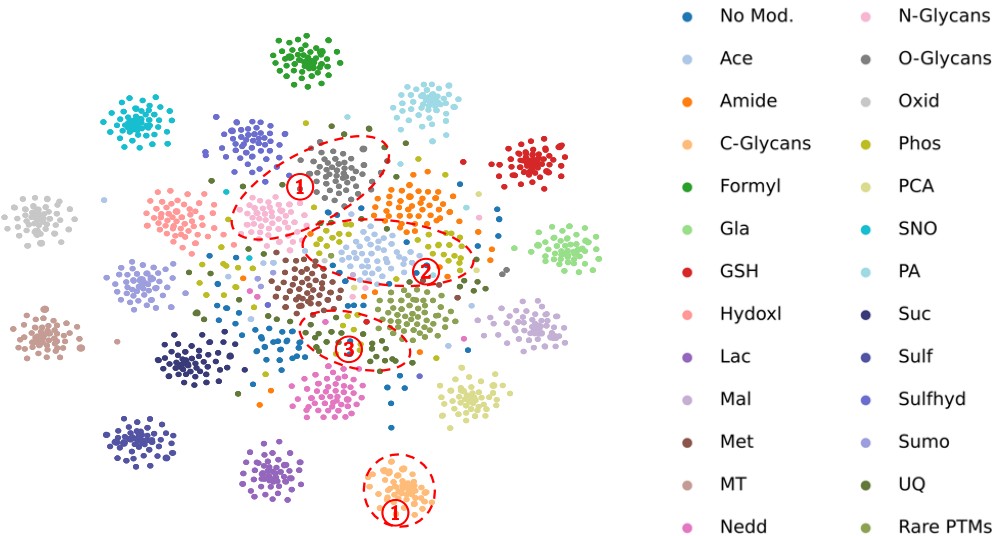

Figure 6: The visualization of the learned codebook.

## 6 CONCLUSION AND LIMITATION

In this study, we focus on PTM prediction, a fundamental challenge in proteomics research. By constructing a large-scale sequence-structure PTM dataset, we established a robust benchmark. Our MeToken harnesses the power of micro-environment tokenization, utilizing uniform codebook loss and temperature-scaled vector quantization to model the interactions within protein micro-environments for predicting long-tail PTM types. Our results demonstrate that MeToken significantly outperforms state-of-the-art approaches, underscoring the importance of incorporating structural data for a comprehensive understanding of PTM sites. The findings offer a pathway to more precise PTM prediction and enhance our understanding of protein functionality.

Despite its strengths, our study has several limitations. While our dataset is a blend of PDB data and AlphaFold database, it relies predominantly on AlphaFold predictions. This dependence may introduce biases inherent to the AlphaFold model. Furthermore, our model currently does not consider interactions between the protein and other molecules, such as enzymes, ligands, or interacting proteins, which play crucial roles in the natural environment of PTMs. The uniform sub-codebook could potentially introduce redundancy for common PTMs by allocating equal capacity across all classes. Addressing these limitations will be a primary focus of our future research endeavors.

## 7 ACKNOWLEDGEMENT

This work was supported by National Science and Technology Major Project (No. 2022ZD0115101), National Natural Science Foundation of China Project (No. 624B2115, No. U21A20427), Project (No. WU2022A009) from the Center of Synthetic Biology and Integrated Bioengineering of Westlake University, Project (No. WU2023C019) from the Westlake University Industries of the Future Research Funding.

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

## A  CODEBOOK SUPPRESSES THE IMPACT OF LONG-TAIL DISTRIBUTION

The "long-tail distribution" issue refers to the phenomenon where a small number of PTM types are highly prevalent, while many others are rare. From an engineering perspective, this issue necessitates a solution that ensures all PTM types, including the rare ones, are adequately represented in the model's learning process. This led us to the development of the sub-codebook strategy in our MeToken model. The engineering solutions can be summarized as follows:

**Uniform representation**  By allocating an identical size sub-codebook to each PTM type, we ensure that rare PTMs have as much representation as others. This uniform distribution of codebooks prevents the model from being dominated by frequent PTMs, promoting a balanced learning process.

**Strong discrimination**  Each sub-codebook is designed to capture distinct patterns specific to its corresponding PTM type. This focused representation allows the model to learn more precise and discriminative features for each PTM type. By ensuring that each PTM type has its own dedicated sub-codebook, the model can better differentiate between various types, improving its ability to identify and predict even the rarest PTMs accurately.

In summary, the sub-codebook strategy addresses the long-tail distribution issue by providing a balanced and discriminative representation for all PTM types, thereby enhancing the model's overall performance and reliability in PTM prediction.

## B  NODE AND EDGE EMBEDDING

Following (Gao et al., 2022), we implement the initial node and edge embeddings for the protein graph with manually constructed structural features. As shown in Table 5, this comprehensive encoding captures the intricate local structural information inherent in each residue of a protein, facilitating a detailed representation crucial for our PTM prediction tasks.

Table 5: The initial node and edge embeddings.

| Level | Feature | Illustration |
|---|---|---|
| Intra-residue Node | Angle | $\big\{ \sin, \cos \big\} \times \big\{ \alpha_i, \beta_i, \gamma_i, \phi_i, \psi_i, \omega_i \big\}$ |
| | Distance | $\Big\{ \mathrm{RBF}(\|\omega_i - \mathrm{C}\alpha_i\|) \,\Big|\, \omega \in \{\mathrm{C}, \mathrm{N}, \mathrm{O}\} \Big\}$ |
| | Direction | $\Big\{ \mathbf{Q}_i^T \, \frac{\omega_i - \mathrm{C}\alpha_i}{\|\omega_i - \mathrm{C}\alpha_i\|} \,\Big|\, \omega \in \{\mathrm{C}, \mathrm{N}, \mathrm{O}\} \Big\}$ |
| Inter-residue Edge | Orientation | $\mathbf{q}(\mathbf{Q}_i^T \mathbf{Q}_j)$ |
| | Distance | $\Big\{ \mathrm{RBF}(\|\omega_i - \omega_j\|) \,\Big|\, j \in \mathcal{N}(i), \omega \in \{\mathrm{C}_\alpha, \mathrm{C}, \mathrm{N}, \mathrm{O}\} \Big\}$ |
| | Direction | $\Big\{ \mathbf{Q}_i^T \, \frac{\omega_i - \omega_j}{\|\omega_i - \omega_j\|} \,\Big|\, j \in \mathcal{N}(i), \omega \in \{\mathrm{C}_\alpha, \mathrm{C}, \mathrm{N}, \mathrm{O}\} \Big\}$ |

Specifically, the node embeddings incorporate a variety of geometric and structural characteristics that provide a comprehensive view of each residue's local environment: (i) angle features, such as $\alpha_i$ (angle for $\mathrm{N}_i$-$\mathrm{C}\alpha_i$-$\mathrm{C}_i$), $\beta_i$ (angle for $\mathrm{C}_{i-1}$-$\mathrm{N}_i$-$\mathrm{C}\alpha_i$), $\gamma_i$ (angle for $\mathrm{C}\alpha_i$-$\mathrm{C}_i$-$\mathrm{N}_{i+1}$), $\phi_i$ (dihedral angle for $\mathrm{N}_i$-$\mathrm{C}\alpha_i$), $\psi_i$ (dihedral angle for $\mathrm{C}$-$\mathrm{C}\alpha_i$), and $\omega_i$ (dihedral angle for $\mathrm{C}_i$-$\mathrm{N}_i$). (ii) distance features, where we employ radial basis functions (RBFs) to transform distances between backbone atoms and the central $\mathrm{C}\alpha_i$ atom into features, capturing the relative positioning of significant structural elements within each residue. (iii) direction features, which are defined by the local coordinate system $\mathbf{Q}_i$ and the normalized vectors pointing from the $\mathrm{C}\alpha_i$ atom to the C, N, O atoms.

Edge embeddings focus on characterizing the interactions between adjacent residues, crucial for understanding how proteins fold and function: (i) orientation features, where $\mathbf{q}(\cdot)$ is the quaternions of relative rotation between the local coordinate systems of two residues. (ii) distance features, which

capture the distances between the backbone atoms of two residues, transformed into radial basis functions. (iii) direction features, which are defined by projecting the normalized vectors between backbone atoms of neighboring residues onto the local coordinate systems $\mathbf{Q}_i$. In our model, the micro-environments of each residue are defined through a comprehensive approach that considers three distinct types of neighbor atom sets. Specifically, the neighborhoods denoted as $\mathcal{N}(i)$ are categorized based on three criteria: sequential order, K-nearest neighbors, and R-radius proximity.

## C DATASET CONSTRUCTION

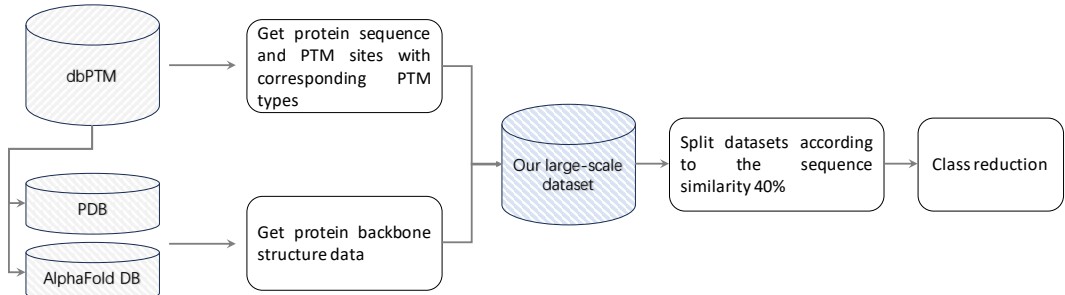

Figure 7: The data processing pipeline for constructing our large-scale PTM dataset.

As illustrated in Figure 7, our dataset construction process was designed to compile a comprehensive and robust resource for investigating post-translational modifications (PTMs) across a wide array of protein types. We began by selecting dbPTM as our foundational sequence-based dataset, which is highly regarded for its extensive catalog of PTM types, encompassing a rich variety of biochemical modifications. In order to enrich this sequence information with structural data, we sourced corresponding structural information from two primary repositories: the Protein Data Bank (PDB) and the AlphaFold database. Our protocol prioritizes experimental data from PDB; however, in instances where such data is unavailable, we resort to high-confidence predictions from AlphaFold, ensuring both accuracy and completeness in our structural dataset representation. Following Dauparas et al. (2022), we introduced Gaussian noise with a mean of zero and a standard deviation of 0.0005 to the atomic coordinates. The addition of this noise simulates small structural fluctuations and uncertainties that may occur in real-world experimental conditions, thereby improving the generalization ability of models trained on our data.

To ensure the integrity of our training and evaluation process and to prevent data leakage, we employed MMseqs2 (Steinegger & Söding, 2017), a highly efficient software suite for handling large biological sequence data through clustering. We clustered the raw data based on sequence similarity, setting a stringent threshold to ensure that proteins in the validation and test sets exhibit no more than 40% similarity to those in the training set. This split provides a reliable measure of performance.

Addressing the challenge of sample imbalance—a common issue in PTM datasets where modified sites are significantly outnumbered by unmodified sites, and some types of modifications are substantially rarer than others. As shown in Figure 8, we consolidated PTM types with fewer than 100 samples into a single 'rare sites' category. We reduced the 73 classes, which included a 'No modification' class to 26 classes. This new classification framework comprises 24 distinct PTM classes, one aggregated class for rare modifications, and one class representing the absence of modification.

In Table 6, we present a comparative analysis between our dataset and other open-sourced PTM datasets. Our dataset contains 1,263,935 PTM sites across 187,812 proteins, offering a detailed exploration of 72 types of modifications. This not only encapsulates a broader diversity of PTM types than typically available but also includes a significantly larger number of protein entries, establishing it as an unparalleled resource in the field. Notably, unlike other datasets such as dbPTM and qPTM, our dataset integrates both sequence and structural data for each protein entry. This integration is crucial for developing more accurate and predictive models of PTM sites, as the structural context often influences where and how modifications occur. Our dataset not only bridges the gap in the availability of comprehensive sequence-structure PTM data but also sets a new standard for the scale and scope of such resources. It enables advanced modeling techniques that can exploit detailed structural contexts alongside sequence information, paving the way for breakthroughs in understanding the complex mechanisms of post-translational modifications.

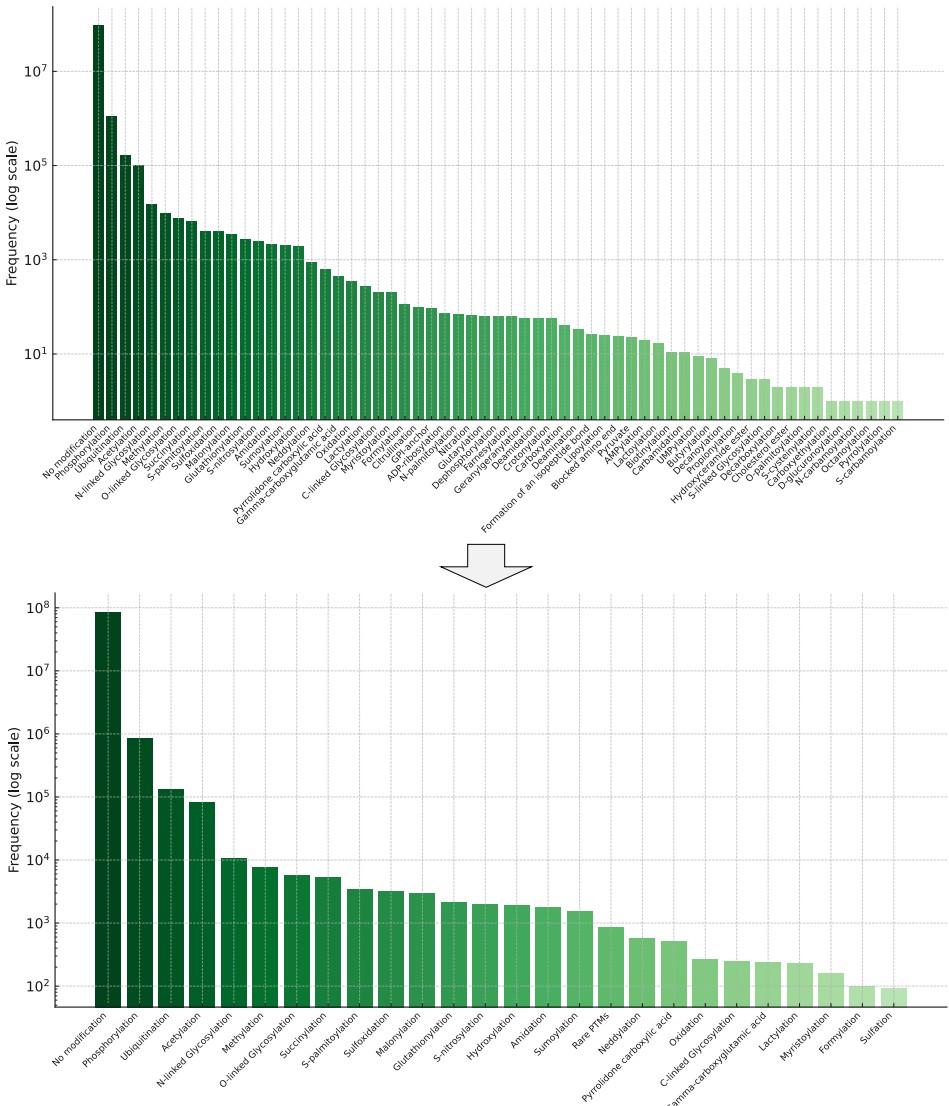

Figure 8: The distribution of PTM classes after class reduction.

The PTMint dataset is a relatively modest collection of 2,477 samples spanning 1,169 proteins with six PTM types. The qPTM dataset offers a broader scope, encompassing 660,030 samples across 40,728 proteins, also classified into the six PTM types. These PTM types are consistent with a subset from our large-scale dataset. Originally, the qPTM dataset lacked comprehensive structural information, which prompted us to enhance it by integrating structural data retrieved from the AlphaFold database.

Table 6: The comparison between open-sourced PTM datasets with our large-scale dataset.

| Database | Number of sites | Number of proteins | Modification types | Sequence | Structure |
|---|---|---|---|---|---|
| PTMint | 2,477 | 1,169 | 6 | ✓ | ✓ |
| PTM Structure | 11,677 | 3,828 | 21 | ✓ | ✓ |
| PRISMOID | 17,145 | 3,919 | 37 | ✓ | ✓ |
| qPTM | 660,030 | 40,728 | 6 | ✓ | ✗ |
| dbPTM | 2,235,000 | 223,678 | 72 | ✓ | ✗ |
| Ours | 1,263,935 | 187,812 | 72 | ✓ | ✓ |

## D   METRICS

To comprehensively evaluate the performance of the PTM prediction task, which we regard as a multi-class classification problem, we adopted a robust suite of metrics. These metrics include accuracy, precision, recall, F1-score, the Matthews Correlation Coefficient (MCC), the Area Under the Receiver Operating Characteristic curve (AUROC), and the Area Under the Precision-Recall Curve (AUPRC). Each of these metrics offers distinct insights into the model's performance, from overall accuracy to the balance between sensitivity and specificity.

**Accuracy**   provides a straightforward measure of the overall correctness of the predictions across all classes. It is calculated as the ratio of correctly predicted observations to the total observations:

$$\text{Accuracy} = \frac{\sum_{i=1}^{N} \mathbb{1}(y_i = \hat{y}_i)}{N}, \tag{8}$$

where $y_i$ is the true PTM type, $\hat{y}_i$ is the predicted PTM type, and $N$ is the total number of instances.

**Precision**   (macro) measures the accuracy of positive predictions. It is especially critical in datasets where false positives carry a significant cost:

$$\text{Precision} = \frac{1}{K} \sum_{k=1}^{K} \frac{\text{TP}_k}{\text{TP}_k + \text{FP}_k}, \tag{9}$$

where $\text{TP}_k$ and $\text{FP}_k$ are the true positives and false positives for the $k$-class, respectively, and $K$ is the total number of classes.

**Recall**   (macro) assesses the model's ability to correctly identify all relevant instances per class:

$$\text{Recall} = \frac{1}{K} \sum_{k=1}^{K} \frac{\text{TP}_k}{\text{TP}_k + \text{FN}_k}, \tag{10}$$

where $\text{FP}_k$ are the false negatives for the $k$-class.

**F1-score**   (macro) is the harmonic mean of precision and recall for each class:

$$\text{F1} - \text{score} = \frac{1}{K} \sum_{k=1}^{K} \frac{2 \times \text{Precision}_k \times \text{Recall}_k}{\text{Precision}_k + \text{Recall}_k}. \tag{11}$$

**Matthews Correlation Coefficient (MCC)**   takes into account true and false positives and negatives and is generally regarded as a balanced measure that can be used even if the classes are of very different sizes:

$$\text{MCC} = \frac{1}{K} \sum_{k=1}^{K} \frac{\text{TP}_k \times \text{TN}_k - \text{FP}_k \times \text{FN}_k}{\sqrt{(\text{TP}_k + \text{FP}_k) \times (\text{TP}_k + \text{FN}_k) \times (\text{TN}_k + \text{FP}_k) \times (\text{TN}_k + \text{FN}_k)}}, \tag{12}$$

**Area Under the Receiver Operating Characteristic curve (AUROC)**   evaluates the trade-off between true positive rate and false positive rate, providing an aggregate measure of performance across all classification thresholds. It reflects the model's ability to discriminate between classes:

$$\text{AUROC} = \frac{1}{K} \sum_{k=1}^{K} \text{AUROC}_k, \tag{13}$$

where $\text{AUROC}_k$ is the AUROC for the $k$-class, defined by integrating the TPR over the range of possible values of FPR:

$$\text{AUROC}_k = \int_0^1 \text{TPR}_k(\text{FPR}_k) \, \text{dFPR}_k, \tag{14}$$

and $\text{TPR}_k = \text{TP}_k/(\text{TP}_k + \text{FN}_k)$ and $\text{FPR}_k = \text{FP}_k/(\text{TN}_k + \text{FP}_k)$ are the true positive rate and false positive rate for the $k$-class, respectively.

**Area Under the Precision-Recall Curve (AUPRC)** assesses the trade-off between Precision and Recall for a given class at various threshold levels, which is particularly useful when dealing with imbalanced datasets:

$$\text{AUPRC} = \frac{1}{K} \sum_{k=1}^{K} \text{AUPRC}_k. \tag{15}$$

where $\text{AUPRC}_k$ is the AUPRC for the $k$-class, defiend as:

$$\text{AUPRC}_k = \int_0^1 \text{Precision}_k(\text{Recall}_k) \, d\text{Recall}_k. \tag{16}$$

## E   IMPLEMENTATION DETAILS

We provide detailed implementation information for the models evaluated in our study. These models are categorized into sequence-based, structure-based, and sequence-structure models, each designed to predict PTM sites based on different data modalities.

### E.1   SEQUENCE-BASED MODELS

**DeepPhos**   As shown in Figure 9, this model (Luo et al., 2019) is composed of densely connected convolutional neural network (DC-CNN) blocks, designed to capture multiple representations of sequences for the final phosphorylation prediction. Specifically, the DC-CNN block operates as a multi-scale pyramid-like merging structure, adept at extracting features from diverse perspectives. Although it aims to predict phosphorylation sites, we adapt the model to predict other PTM types by modifying the output layer to accommodate the desired PTM classes.

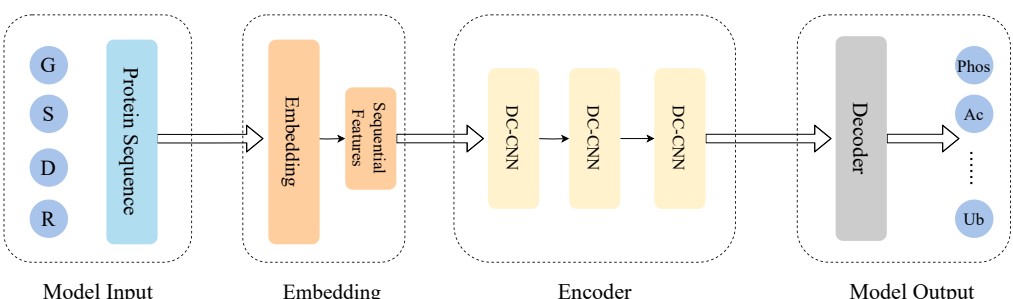

Figure 9: The model architecture of DeepPhos.

**EMBER**   As shown in Figure 10, EMBER(Kirchoff & Gomez, 2022) employs a modified Siamese neural network tailored for multi-label prediction tasks to generate high-dimensional embeddings of motif vectors. Additionally, it uses one-hot encoded motif sequences and leverages these two distinct representations as dual inputs into the classifier.

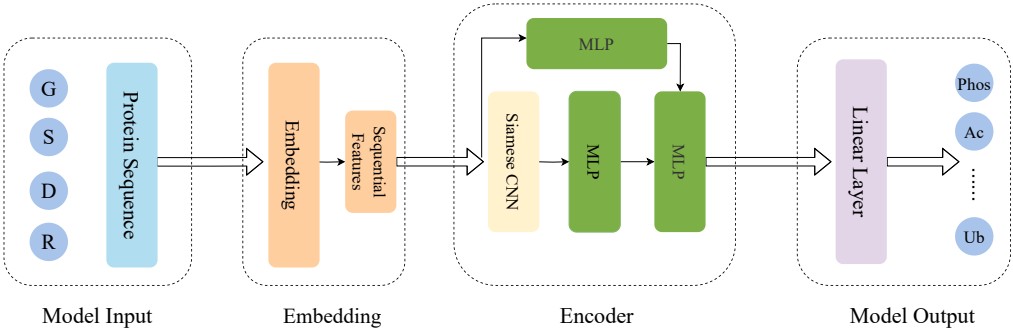

Figure 10: The model architecture of EMBER.

**MusiteDeep** As shown in Figure 11, Multi-layer CNN is used as the feature extractor but no pooling layers are used. The last hidden state of multi-layer CNN is copied twice, where one directly inputs into the attention mechanism and the other first trans-positioned and then inputs into another attention mechanism. The output of the two attention mechanisms is combined and input into the fully connected neural network layers. The final layer is a single layer with the softmax output.

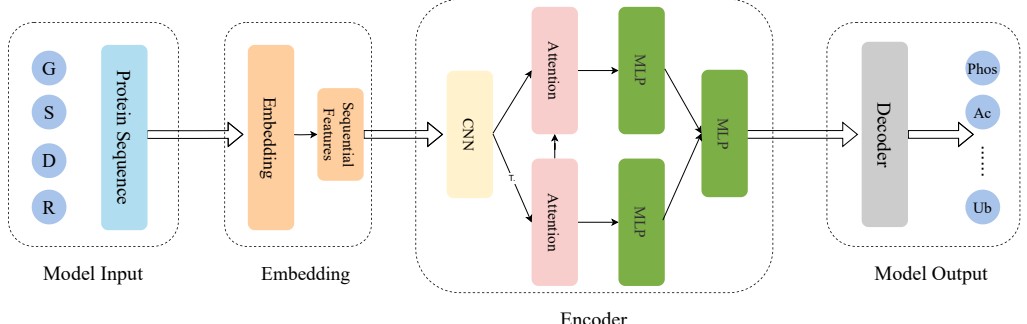

Figure 11: The model architecture of MusiteDeep.

**ESM** As shown in Figure 12, we utilize the pretrained model from ESM (Lin et al., 2023) to obtain embeddings of protein sequences. These embeddings serve as a rich representation of the protein data, capturing intricate evolutionary patterns. Following this, an MLP is employed as the classifier.

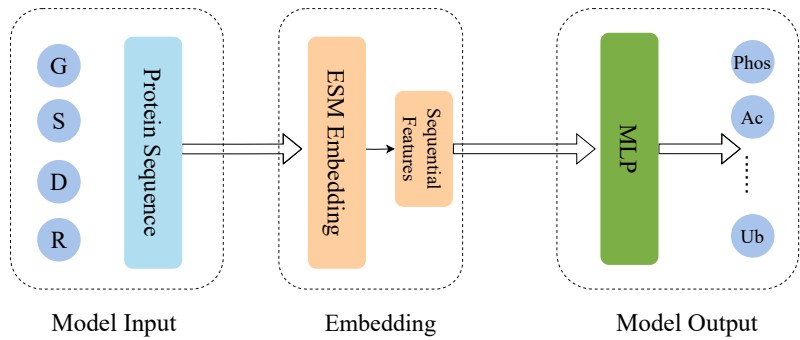

Figure 12: The model architecture of ESM.

### E.2 STRUCTURE-BASED MODELS

**StructGNN** As shown in Figure 13, StructGNN (Ingraham et al., 2019) is a traditional protein design model that has been adapted in this work to predict residue-level post-translational modification (PTM) sites. The model begins by obtaining structural features through a conventional structure embedding process. These features capture the intricate three-dimensional configurations of protein structures. Following this, the model employs layers of a Message Passing Neural Network, which iteratively refines the structural representations by aggregating information across the network.

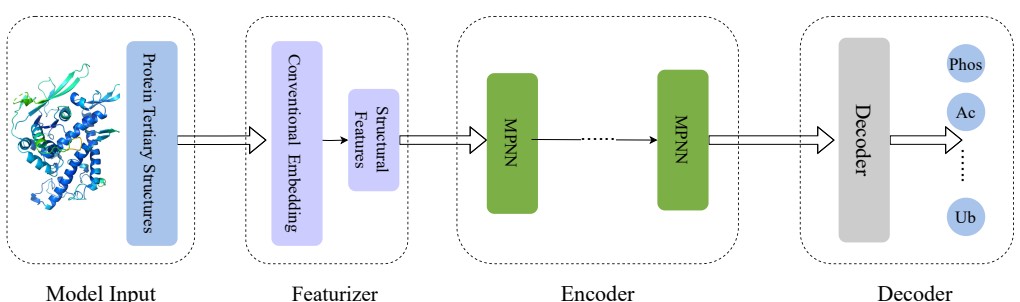

Figure 13: The model architecture of StructGNN.

**GraphTrans**   Figure 14 shows that GraphTrans (Ingraham et al., 2019) is a Transformer-based model that replaces StructGNN's MPNN layers with attention mechanisms. This modification allows the model to capture long-range dependencies and intricate interactions within the protein structure more effectively.

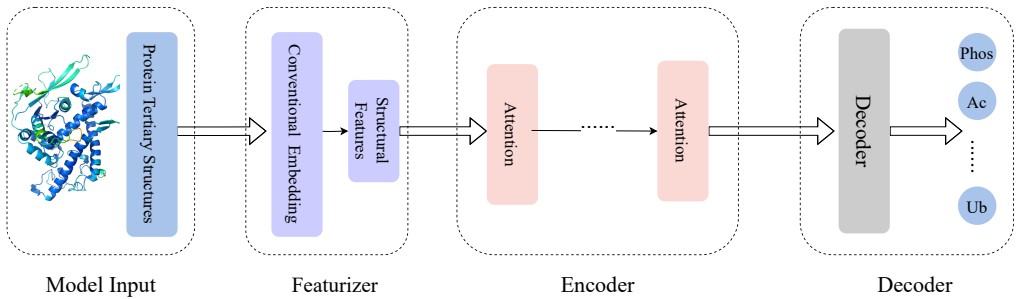

Figure 14: The model architecture of GraphTrans

**GVP**   GVP (Jing et al., 2020) introduce geometric vector perceptrons, which extend standard dense layers to operate on collections of Euclidean vectors. Graph neural networks equipped with such layers are able to perform both geometric and relational reasoning on efficient representations of macromolecules.

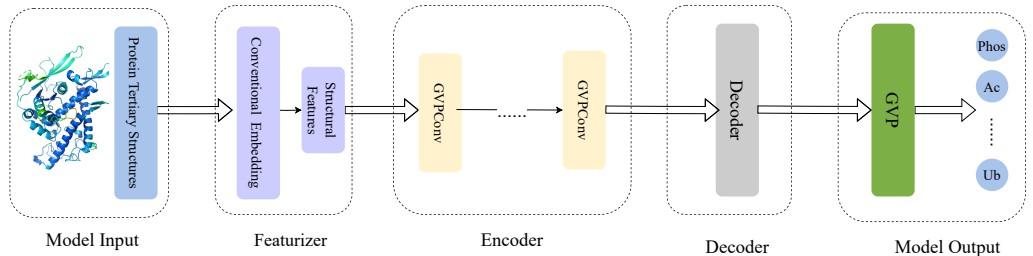

Figure 15: The model architecture of GVP.

**PiFold**   This model extracts features of protein structures and employs a sophisticated encoder-decoder architecture. The featurizer module leverages virtual atoms to enhance the representation of spatial information, capturing critical characteristics such as distances, angles, and directions. PiGNN that tailored for protein structure representation to encode structural features and MLP as a decoder.

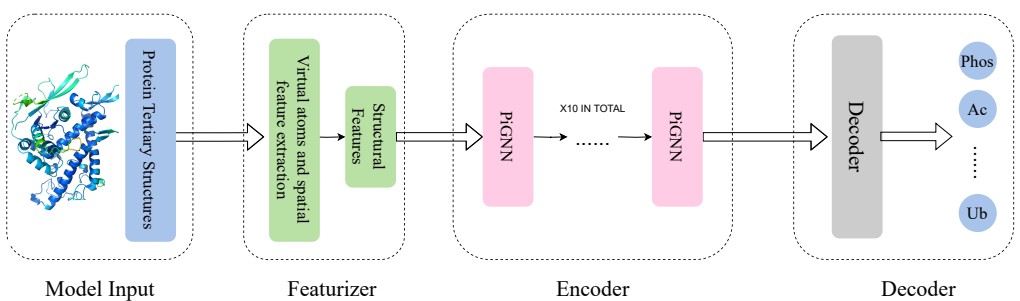

Figure 16: The model architecture of PiFold.

### E.3   SEQUENCE-STRUCTURE MODELS

**GearNet**   As depicted in Figure 17, GearNet (Zhang et al., 2022)—the Geometry-Aware Relational Graph Neural Network—is a simple yet highly effective structure-based protein encoder. GearNet

encodes spatial information by incorporating various types of sequential and structural edges into the protein residue graphs. It then performs relational message passing, allowing the model to capture and integrate complex spatial relationships within the protein structure.

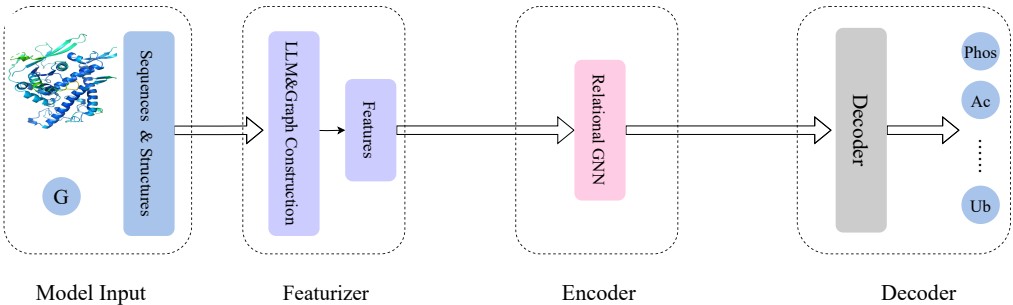

Figure 17: The model architecture of GearNet.

**CDConv**    As illustrated in Figure 18, the Continuous-Discrete Convolution (CDConv) is introduced to simultaneously model the geometry and sequence structures of proteins using distinct approaches for their regular and irregular characteristics. CDConv employs independent learnable weights to handle regular sequential displacements within the protein sequence. For the irregular geometric displacements, it directly encodes these variations to capture the complex spatial relationships inherent in the protein's three-dimensional structure.

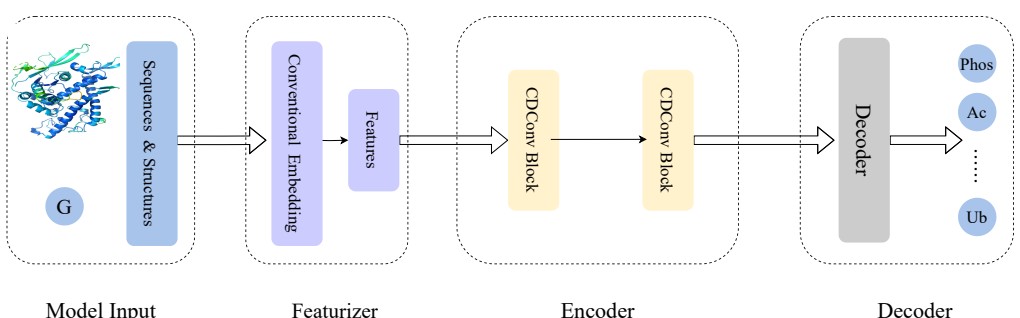

Figure 18: The model architecture of CDConv.

**MIND**    As illustrated in Figure 19, the MIND model employs a dual-path approach to analyze protein data. One path leverages LSTM networks combined with attention mechanisms to scrutinize the sequence information. The other path constructs the protein structure as a graph and utilizes graph attention mechanisms to analyze the structural data. A MLP then integrates the information from both paths, functioning as a classifier to make final predictions.

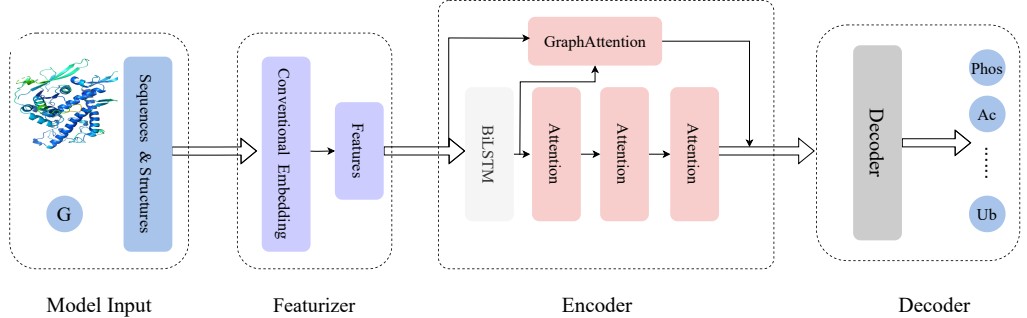

Figure 19: The model architecture of MIND.

# F Supplementary experiments

## F.1 ESM-3 and SaProt baselines

In our investigation, we also examined the performance of the ESM-3 (Hayes et al., 2024) model and the SaProt (Su et al., 2023) model, as shown in Table 7. Surprisingly, our results indicated that fine-tuning ESM-2 outperforms fine-tuning ESM-3 in this context. Previous research (West-Roberts et al., 2024) has suggested that ESM-3 does not consistently outperform ESM-2, highlighting an intriguing nuance in model performance. One potential explanation for this phenomenon is that ESM-3 incorporates considerations of sequence, structure, and function, which may introduce a level of complexity that does not always lead to enhanced performance for specific tasks like PTM prediction. Moreover, the SaProt model, which leverages Foldseek's discrete 3D tokens, demonstrates slightly better performance than ESM-2 across most metrics. This suggests that incorporating structure-aware representations can be beneficial for PTM prediction. However, the performance improvement is relatively minor, likely due to the inherent limitations of Foldseek's tokenization scheme. Specifically, Foldseek's discrete 3D tokens are limited to 20 types, which may fail to capture the level of fine-grained structural detail required for modeling the nuanced micro-environments around PTM sites.

Table 7: Performance comparison between ESM-2 and ESM-3 models.

| Method | F1 | Acc | Pre | Rec | MCC | AUROC | AUPRC |
|--------|------|------|------|------|------|-------|-------|
| ESM2 | 43.94 | 89.75 | 55.67 | 42.15 | 75.83 | 91.52 | 47.31 |
| ESM3 | 40.65 | 87.48 | 45.72 | 38.67 | 70.31 | 89.91 | 41.36 |
| SaProt | 45.18 | 89.87 | 57.23 | 43.14 | 75.66 | 93.66 | 48.81 |

## F.2 Other sequence-structure baselines

We also conducted experiments using additional sequence-structure models to further contextualize our findings, as shown in Table 8. The STEPS model (Chen et al., 2023) structures data as a graph and utilizes sequence language models for embedding sequences, focusing on maximizing the mutual information between sequence and structure through a pseudo bi-level optimization approach. However, unlike MeToken, which is specifically designed for residue-level tasks, STEPS is tailored primarily for protein-level applications, limiting its effectiveness in our targeted analysis. Similarly, MPRL (Nguyen & Hy, 2024) adopts a multimodal framework that integrates ESM-2 for sequence embedding, alongside VGAE and point cloud techniques for structure embedding, culminating in a fusion module that generates a comprehensive representative embedding. While MPRL demonstrates strong performance in protein-level tasks, it struggles to adequately address the nuances of residue-level tasks, revealing a gap in its applicability. The results indicate that both STEPS and MPRL, while effective in their respective domains, demonstrate limitations when applied to residue-level tasks, further emphasizing the unique capabilities of MeToken in this PTM prediction.

Table 8: Performance metrics for STEPS and MPRL models.

| Method | F1 | Acc | Pre | Rec | MCC | AUROC | AUPRC |
|--------|------|------|------|------|------|-------|-------|
| STEPS | 15.96 | 66.32 | 13.26 | 20.04 | 2.80 | 41.72 | 19.69 |
| MPRL | 19.04 | 90.39 | 18.08 | 20.11 | 5.02 | 52.68 | 21.55 |

## F.3 The reliability of AlphaFold2's predicted data

To assess the reliability of our large-scale dataset, we conducted a thorough validation analysis in Table 9. The combination of AlphaFold2 predictions and PDB data produced the highest performance metrics, highlighting the importance of combining predicted and experimental structures. Specifically, the AF2-only dataset demonstrated significantly lower performance, revealing the limitations inherent in relying solely on predictions. In contrast, the PDB-only dataset outperformed AF2, yet it still did not match the performance of the hybrid AF2+PDB dataset. This indicates that while experimental data plays a critical role, the extensive scale of AF2 predictions substantially enhances overall results.

Table 9: Performance metrics across different datasets.

| Dataset | F1 | Acc | Pre | Rec | MCC | AUROC | AUPRC |
|---------|------|------|------|------|------|------|------|
| AF2+PDB | 50.40 | 89.91 | 58.08 | 47.93 | 76.03 | 92.20 | 51.26 |
| AF2 | 11.74 | 87.23 | 10.29 | 14.80 | 54.54 | 41.79 | 12.95 |
| PDB | 41.31 | 73.39 | 40.96 | 47.10 | 52.25 | 91.07 | 40.87 |

These findings suggest that, although AF2 data may introduce certain biases, its large scale contributes positively to performance. Ultimately, the most effective outcomes arise from integrating both predicted and experimental data, achieving an optimal balance between scale and accuracy. This analysis reinforces the conclusion that the integration of AF2 predictions with PDB data not only mitigates potential biases but also maximizes predictive power, thereby enhancing the reliability of our large-scale dataset and its applicability in protein research.

### F.4 STATISTICS ABOUT CODEBOOK DISTRIBUTION

To illustrate the distribution of tokens within our codebook, we examine a specific token with index 2588. As depicted in Figure 20, the most prevalent amino acids associated with this token are valine (Val, V) and leucine (Leu, L), both of which are hydrophobic amino acids characterized by similar structural properties. The other three amino acids in the distribution are also hydrophobic, reinforcing this trend. This observation indicates that our model has effectively learned and captured significant physicochemical features of amino acids within the codebook. Such a finding enhances the credibility and interpretability of our model, demonstrating its ability to reflect meaningful biological properties in its decent representations.

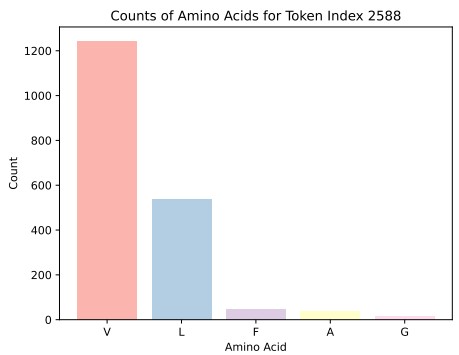

Figure 20: The corresponding amnio acid distribution of the token 2588.

## G ABBREVIATIONS OF PTMS

Due to space constraints, we have used abbreviations in some places in the main text for PTMs. Below is the table of abbreviations and their corresponding full names.

## H UMAP VISUALIZATION OF METOKEN CODEBOOK

UMAP visualization captures similar biological concept to previous tSNE visualization, though with some difference.

- Acetylation, ubiquitination and phosphorylation has overlap areas and a short distance, which in some case proves the relationship, similarity and crosstalks among these types
- N-linked Glycosylation has a much shorter distance with O-linked Glycosylation than C-linked Glycosylation, echoing the ground truth of the bioligical process of these three modifications, just as shown in the text.
- UMAP better captures the interior features in the codebook embedding
- UMAP reduces the distance between similar classes and increases the distance between dissimilar classes.

## I VISUALIZATION OF THE ABILITY TO DISTINGUISH SIMILAR CLASSES

Accurately distinguishing between similar post-translational modification (PTM) classes poses a significant challenge due to their overlapping biochemical and structural characteristics. To evaluate

Table 10: Abbreviations and corresponding names of PTMs.

| Abbreviation | PTM types |
| --- | --- |
| Ac | Acetylation |
| Amide | Amidation |
| C-Glycans | C-linked Glycosylation |
| Formyl | Formylation |
| Gla | Gamma-carboxyglutamic acid |
| GSH | Glutathionylation |
| Hydroxyl | Hydroxylation |
| Lac | Lactylation |
| Mal | Malonylation |
| Met | Methylation |
| MT | Myristoylation |
| Nedd | Neddylation |
| N-Glycans | N-linked Glycosylation |
| O-Glycan | O-linked Glycosylation |
| Oxid | Oxidation |
| Phos | Phosphorylation |
| PCA | Pyrrolidone carboxylic acid |
| SNO | S-nitrosylation |
| PA | S-palmitoylation |
| Suc | Succinylation |
| Sulf | Sulfation |
| Sulfhyd | Sulfoxidation |
| Sumo | Sumoylation |
| Ub | Ubiquitination |

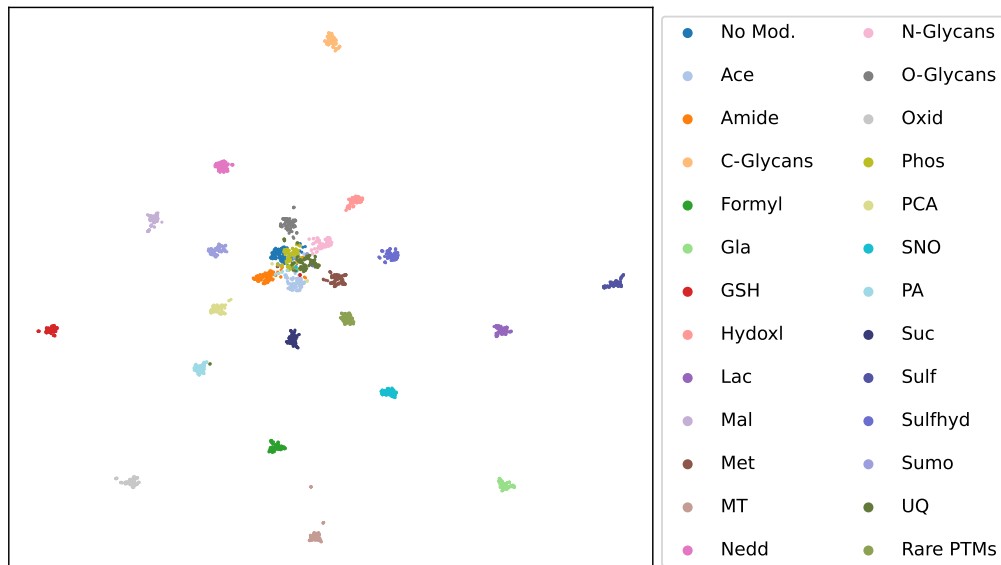

Figure 21: UMAP Visualization of MeToken codebook

our model's capacity to differentiate between such classes, we use SUMOylation and ubiquitination as a representative example. Figure 22 shows a t-SNE visualization of the learned embeddings for SUMOylation and ubiquitination sites. Although some overlap exists between the two distributions, the SUMOylation embeddings are primarily concentrated in a distinct subregion of the ubiquitination

distribution. This spatial separation demonstrates that the embeddings produced by our model effectively capture meaningful features that differentiate these two classes, even in the presence of significant similarity.

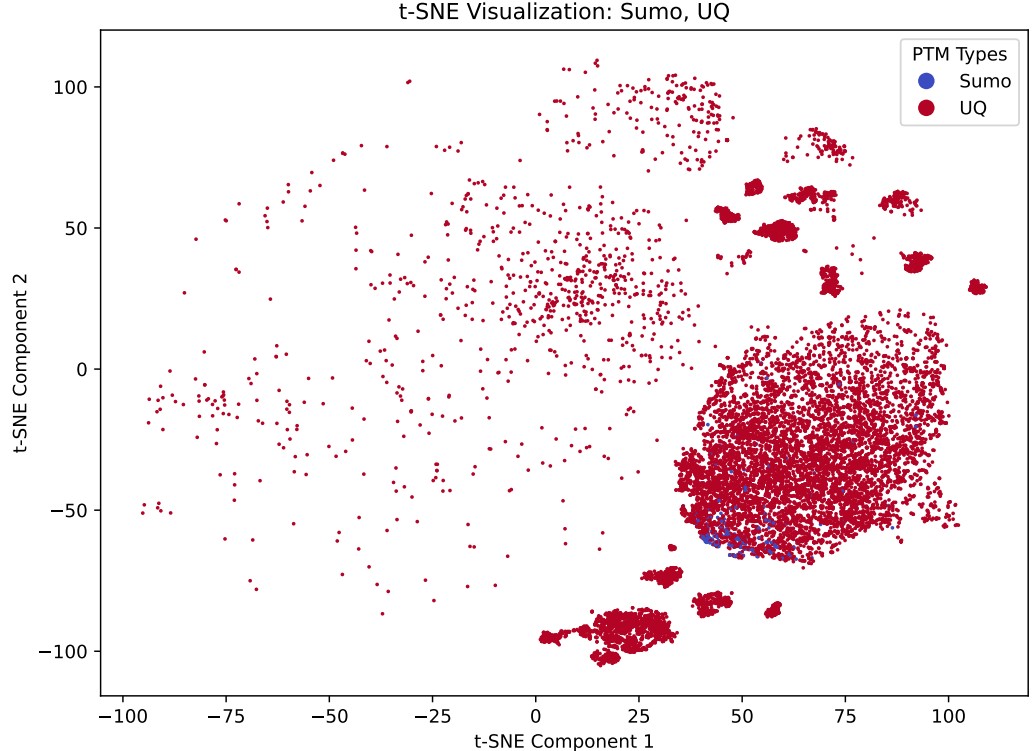

Figure 22: t-SNE visualization of SUMOylation and ubiquitination

To quantitatively assess the separability of the two classes, we trained a Support Vector Machine (SVM) classifier on the embeddings generated by our model. The SVM achieved an accuracy of 0.9881 and an F1-score of 0.99, indicating that the embeddings are highly discriminative. This result highlights the robustness of our model in distinguishing between SUMOylation and ubiquitination, which are notoriously difficult to separate due to their shared preference for lysine residues and overlapping structural contexts.

These findings illustrate the strength of our micro-environment tokenization approach in encoding subtle yet crucial differences between similar PTM types. The ability to classify SUMOylation and ubiquitination with such high accuracy underscores the utility of our model in resolving classification challenges among closely related PTM classes.

## J  PTMs Categorization and MeToken's Performance

PTMs can be broadly categorized into four types based on the nature of the modifications: chemical groups, proteins/peptides, complex molecules, and cleavage events. Each category has unique structural and sequence characteristics, which influence the predictive capability of computational models like MeToken. Below, we summarize these categories and discuss how MeToken is designed to capture their specific features:

- Chemical group modifications (like phosphorylation) often depend on specific sequence motifs and can occur on exposed residues, which are easier for the model to identify through sequence context and local structural features.

- Protein/peptide modifications (e.g., ubiquitylation) typically require a spatially accessible lysine residue, along with structural features that might allow for binding of modifying enzymes. MeToken's structural context encoding is particularly beneficial here.

- Complex molecules (e.g., glycosylation) generally have specific sequence motifs and often require certain three-dimensional accessibility, which the model can learn through spatial tokenization.

- Cleavage events (like proteolysis) may have broader sequence motifs but are often associated with regions of the protein that are structurally exposed or flexible, properties that are encoded in the model through local structural information. **Since our model is built on the consistency of the lengths of structural data, sequence data, and PTM modification data, we do not consider this type in our dataset.

Table 11 summarizes the performance of MeToken on different PTM categories. MeToken performs best on chemical group modifications, achieving the highest scores across nearly all metrics. This is likely because these modifications often depend on well-defined sequence motifs, which the model can easily learn. Additionally, their occurrence on solvent-exposed residues provides consistent structural cues that enhance prediction accuracy. Performance on protein/peptide modifications, such as ubiquitylation, is lower across most metrics. Predictions for complex molecule modifications, such as glycosylation, achieve intermediate performance. It suggests that MeToken can capture some key features, such as sequence motifs and spatial accessibility.

Table 11: Performance analysis across PTM categories.

| Method | F1 | Acc | Pre | Rec | MCC | AUROC | AUPRC |
|---|---|---|---|---|---|---|---|
| Chem Groups | 0.6165 | 0.9533 | 0.6968 | 0.5809 | 0.7640 | 0.9902 | 0.6538 |
| Protein/Peptide | 0.3798 | 0.6903 | 0.5579 | 0.3038 | 0.0477 | 0.6851 | 0.4650 |
| Complex Molecules | 0.3470 | 0.7071 | 0.6667 | 0.3344 | 0.4850 | 0.6654 | 0.6643 |

## K    COMPARISON OF MeToken WITH OTHER TOKENIZATION IN MOLECULAR MODELING

**FoldToken**    The main contribution of FoldToken Gao et al. (2024) is its SoftCVQ (Soft Conditional Vector Quantization), which aligns the embedding induced by the binary code index with a smooth gradient using an adjustable temperature parameter. Our approach shares this similarity in that we also utilize an adjustable temperature to ensure smooth gradient optimization during training. However, there are key differences in the context and application of these techniques.

While FoldToken is designed for protein backbone generation and requires a highly expressive latent space to effectively reconstruct detailed 3D structures, our work focuses on protein PTM prediction, aiming to learn a compact latent space that captures relevant sequence-structure relationships specific to PTM sites. To achieve this, we introduce a uniform sub-codebook strategy, which helps to more effectively model PTM types by partitioning the latent space into smaller, specialized sub-codebooks for each PTM subtype. This is particularly useful in the context of PTM prediction, where we aim to capture the distinct modification patterns and functional relationships between amino acids at modification sites. Furthermore, while FoldToken uses a large-scale binary codebook (codebook size = 65536) to allow for detailed protein structure reconstruction, we focus on designing a smaller (codebook size = $26 \times 128 = 3328$), task-specific latent space tailored to PTM representation learning. The sub-codebook strategy in our model ensures that the embeddings focus on the most relevant features for PTM prediction, improving model efficiency and interpretability. Additionally, merely relying on smooth gradient optimization through adjustable temperature is not sufficient for effective PTM prediction, which is why we also incorporate the sub-codebook approach to explicitly guide the model toward the most important features for PTM classification.

**MAPE-PPI**    The MAPE-PPI model Wu et al. (2024) is designed to predict protein-protein interactions (PPIs) by employing a multi-scale attention mechanism to capture the intricate relationships between protein sequences and structures. While MAPE-PPI specifically targets PPI prediction, our

work is focused on PTM prediction, a task that demands a distinct set of features and representations. The tokenization process in MAPE-PPI emphasizes aggregating information across multiple scales to capture hierarchical relationships, whereas our MeToken approach addresses the challenges of the long-tail distribution inherent to PTM datasets by employing a sub-codebook strategy to ensure even rare PTMs are adequately represented.

Furthermore, MAPE-PPI utilizes vanilla VQ, which performs hard assignment of input features to codebook vectors. In contrast, our method incorporates a temperature-scaled vector quantization mechanism, enabling a smoother transition between exploratory and deterministic quantization during training. This allows for more robust learning of complex micro-environmental features critical for PTM prediction. These methodological differences highlight how our MeToken model is tailored to the unique challenges of PTM site prediction, as opposed to MAPE-PPI's focus on PPIs.

