# OpenReview forum: "MeToken: Uniform Micro-environment Token Boosts Post-Translational Modification Prediction"
_ICLR.cc/2025/Conference — ICLR 2025 Poster_

### Official Review · Reviewer_GBsm · 2024-11-01

**Soundness:** 3
**Presentation:** 3
**Contribution:** 3
**Rating:** 5
**Confidence:** 5

**Summary:**

PTM prediction is a very important topic in bioinformatics. This study introduces the MeToken model to consider the micro-environment of amino acids for PTM prediction. The proposed method has shown promising results in performance evaluation tests.

**Strengths:**

The authors incorporated the protein structural data to enhance the prediction, which is good as structural information could be better presented the functional-related information.

**Weaknesses:**

Some descriptions are not very accurate, and the progress of some existing studies is ignored.

**Questions:**

1. PTMs can be summarised into four types according to the modification types: chemical groups (e.g. Phosphorylation, Acetylation), proteins/peptides (e.g. Ubiquitylation, Sumoylation), complex molecules (e.g. glycosylation), and cleavage (proteolysis). These four types could have different structural preferences. How can the proposed model capture the differences between them? Which types work well using the proposed method?
2. The description "Existing computational approaches predominantly focus on protein sequences to predict PTM sites" in the Abstract is not accurate. There are many graphs and structural-aware methods that have been developed and published.
3. The authors are suggested to provide some biological case studies to interpret the micro-environment tokenization related to the characteristics of PTMs in the biological context.
4. The source codes and datasets should be publicly available upon acceptance.

---

> ### Author Response · Authors · 2024-11-17
> **Response to Reviewer GBsm (1/2)**
>
> Dear Reviewer GBsm,
>
> Thank you for your thoughtful feedback and constructive suggestions. We appreciate the opportunity to address your questions and comments, and we provide detailed responses below. Your insights have highlighted several areas where we can improve the clarity and depth of our manuscript, and we are committed to incorporating these changes in the revised version.
>
> **Q1** PTMs can be summarised into four types according to the modification types: chemical groups (e.g. Phosphorylation, Acetylation), proteins/peptides (e.g. Ubiquitylation, Sumoylation), complex molecules (e.g. glycosylation), and cleavage (proteolysis). These four types could have different structural preferences. How can the proposed model capture the differences between them? Which types work well using the proposed method?
>
> **A1** This is an insightful question, and we appreciate the opportunity to clarify how our model captures these differences. MeToken’s micro-environment tokenization approach is designed to capture both the local sequence motifs and spatial context around each residue, which should theoretically help distinguish PTMs based on their structural preferences:
> - Chemical group modifications (like phosphorylation) often depend on specific sequence motifs and can occur on exposed residues, which are easier for the model to identify through sequence context and local structural features.
> - Protein/peptide modifications (e.g., ubiquitylation) typically require a spatially accessible lysine residue, along with structural features that might allow for binding of modifying enzymes. MeToken’s structural context encoding is particularly beneficial here.
> - Complex molecules (e.g., glycosylation) generally have specific sequence motifs and often require certain three-dimensional accessibility, which the model can learn through spatial tokenization.
> - Cleavage events (like proteolysis) may have broader sequence motifs but are often associated with regions of the protein that are structurally exposed or flexible, properties that are encoded in the model through local structural information. **Since our model is built on the consistency of the lengths of structural data, sequence data, and PTM modification data, we do not consider this type in our dataset.**
>
> | Method | F1 Score | Accuracy | Precision | Recall |  MCC Score | AUROC  | AUPRC  |
> | ----------------- | -------- | --------- | ------ | -------- | --------- | ------ | ------ |
> | Chem Groups       | 0.6165 | 0.9533 | 0.6968 | 0.5809 |  0.7640 | 0.9902 | 0.6538 |
> | Protein/Peptide   |  0.3798 | 0.6903 | 0.5579 | 0.3038 | 0.0477 | 0.6851 | 0.4650 |
> | Complex Molecules | 0.3470 | 0.7071 | 0.6667 | 0.3344 |  0.4850 | 0.6654 | 0.6643 |
>
> MeToken performs best on chemical group modifications, achieving the highest F1 score. These modifications often have strong sequence motifs and occur on solvent-accessible residues, making them easier for the model to identify. The model achieves moderate performance for complex molecule modificationsm due to the complex and varied structural contexts in which glycosylation occurs.
>
> **Q2** The description "Existing computational approaches predominantly focus on protein sequences to predict PTM sites" in the Abstract is not accurate. There are many graphs and structural-aware methods that have been developed and published.
>
> **A2** Thank you for this important observation. If you are aware of additional studies that would further strengthen our discussion, we would greatly appreciate your recommendations. We are eager to learn about and incorporate any relevant advancements that may enhance the context and positioning of our work.

---

> > ### Author Response · Authors · 2024-11-17
> > **Response to Reviewer GBsm (2/2)**
> >
> > **Q3** The authors are suggested to provide some biological case studies to interpret the micro-environment tokenization related to the characteristics of PTMs in the biological context.
> >
> > **A3** Thank you for this insightful suggestion. We agree that providing biological case studies would enhance the interpretability and biological relevance of our work.
> >
> > As illustrated in Figure 6 of the manuscript, we used t-SNE to visualize the learned code embeddings from MeToken and examined the relationships between PTMs in the latent space. This analysis reveals biologically meaningful patterns:
> >
> > - Relationship Between Different Types of Glycosylation: C-glycosylation is positioned notably far from N- and O-glycosylation. This distinction aligns with the fundamental biochemical differences between these types of glycosylation. N- and O-glycosylation involve the attachment of sugars to nitrogen (N) or oxygen (O) atoms, respectively, whereas C-glycosylation involves the formation of a covalent bond between a sugar molecule and a carbon atom. This difference is reflected in their distinct clustering in the latent space, showcasing MeToken's ability to distinguish PTMs based on underlying structural and biochemical properties.
> > - Competitive Modifications Between Phosphorylation and Acetylation: Phosphorylation and acetylation sites are scattered and sometimes overlap in the t-SNE plot, reflecting their competitive nature. This is consistent with experimental findings that these two PTMs often compete for modification sites on serine, threonine, and tyrosine residues. MeToken captures this competitive interplay by placing these PTMs in partially overlapping clusters, indicating shared structural preferences or enzymatic contexts.
> > - Distribution of Phosphorylation and Ubiquitination Sites: The dispersed distribution of phosphorylation and ubiquitination sites in the latent space indicates that these PTMs are heterogeneous in their structural contexts. This fragmented pattern aligns with prior studies, which emphasize the diversity of structural environments that can accommodate phosphorylation (frequently surface-exposed regions) and ubiquitination (often requiring accessible lysine residues).
> >
> > These observations highlight MeToken's ability to capture the functional and structural diversity inherent in these modifications.
> >
> > To further interpret the micro-environment tokenization, we examined a specific token within our codebook (index 2588) as an example. This analysis, shown in **Figure 20 of the Appendix F.4**, demonstrates the ability of MeToken to capture key physicochemical properties:
> > - The most prevalent amino acids associated with token 2588 are valine (Val, V) and leucine (Leu, L), both of which are hydrophobic residues. The distribution also includes other hydrophobic residues, indicating that this token captures the hydrophobic character of certain micro-environments. This token likely corresponds to regions of proteins with non-polar side chains, which are often buried in the protein core or involved in hydrophobic interactions.
> > - Certain PTMs, such as lipidation or ubiquitination, may favor hydrophobic environments due to their interaction with non-polar regions of proteins. The fact that this token groups residues with similar physicochemical properties suggests that MeToken effectively encodes structural features relevant to the function of PTMs.
> >
> > These findings suggest that MeToken is not merely learning shallow patterns but is instead capturing meaningful biological features that are relevant to the structural and functional contexts of PTMs.
> >
> > **Q4** The source codes and datasets should be publicly available upon acceptance.
> >
> > **A4** We fully agree with the importance of making our code and datasets publicly available to facilitate reproducibility and further research. Upon acceptance, we will release our code, model weights, and the compiled sequence-structure PTM dataset on a public repository.
> >
> > To ensure anonymity during the review process, we have provided a temporary, anonymized link to the code for reviewers to verify our implementation: https://anonymous.4open.science/r/MeToken-7E64/. This temporary repository includes the essential components of our codebase. We will make the full, finalized repository publicly accessible upon acceptance.
> >
> > ----
> >
> > We would like to sincerely thank you for your thoughtful and constructive feedback during the initial review process. Your comments have been instrumental in improving the quality and clarity of our work. We have carefully addressed all of your concerns to the best of our ability and hope that our responses and revisions adequately resolve the issues you raised.
> >
> > If you find our efforts and the resulting improvements satisfactory, we would greatly appreciate it if you could revisit and reconsider your assessment of our submission. Your input is invaluable, and we remain fully committed to delivering a strong and impactful contribution to the field.

---

> > ### Comment · Reviewer_GBsm · 2024-11-22
> > **Graphs and structural-aware methods for PTM site prediction**
> >
> > Q2 is to suggest that the author should be cautious about the description of existing research, and the literature review should be the authors' own responsibility. However, I still helped the authors search for some relevant literature, which proves that there are more and more studies on PTM site prediction using graph neural networks and structural perception for the author to choose from:
> >
> > 1. https://doi.org/10.1109/BIBM55620.2022.9995153
> > 2. https://doi.org/10.1016/j.crmeth.2023.100430
> > 3. https://doi.org/10.1093/bioinformatics/btae675
> > 4. https://doi.org/10.1016/j.xpro.2023.102682
> > 5. https://doi.org/10.1016/j.eswa.2024.124770
> > 6. https://doi.org/10.1371/journal.pcbi.1011939
> > 7. https://doig.org/10.1109/ACCESS.2024.3427792
> >
> > Please note that none of the above papers are the reviewer's papers.

---

> > > ### Author Response · Authors · 2024-11-22
> > >
> > > Thank you very much for your valuable feedback and for providing a comprehensive list of relevant literature on PTM prediction using graph neural networks and structure-aware approaches. We appreciate the time and effort you have taken to assist us in ensuring our work is grounded within the broader research context.
> > >
> > > In summary, **MIND-S [2,4] has already been included in our baseline and referred to as MIND for simplicity.** While we believe our initial review captured many of the major existing works, we acknowledge that the field of PTM prediction has been advancing rapidly, with several highly relevant papers published in this year:
> > > - **DeepPCT [3] published on 21 November 2024 (just yesterday)**;
> > > - **WPTCMN/PTCMN [5] published on 24 July 2024**;
> > > - **Ertelt et al. [6] published on March 14, 2024**;
> > > - **PhosHSGN [7] published on July 29, 2024**.
> > >
> > >  We also note that some of the suggested references were not available at the time of our submission or are extremely recent contributions. **Nevertheless, we are grateful for your suggestions and have carefully analyzed the provided references.** Here, we summarize how they relate to our work and highlight key distinctions:
> > >
> > > GraphUbiquSite [1] uses a capsule neural network rather than an attention-based module for feature embedding. While its approach to leveraging AlphaFold Database (AFDB) for structure prediction aligns with ours, there are key differences. MeToken prioritizes experimental data (e.g., from PDB) for structural information and only resorts to AlphaFold-predicted structures when experimental data is unavailable. Furthermore, GraphUbiquSite constructs residue graphs based on distance thresholds alone, whereas MeToken employs a more sophisticated graph construction strategy that combines K-nearest neighbors, sequential neighbors, and R-radius neighbors. This multi-faceted graph representation allows MeToken to better capture both local and global residue interactions. Another distinction lies in how embeddings are learned; GraphUbiquSite does not use learned sequence representations and instead relies on pre-trained embeddings from language models, whereas MeToken learns its sequence representations directly, tailored specifically for PTM prediction.
> > >
> > > **DeepPCT [3] tackles a different task than MeToken, focusing on PTM crosstalk prediction.** While MeToken predicts the PTM type for a single site, DeepPCT aims to determine the relationship between two PTM sites using sequence, structure, and functional context. Additionally, the structural feature extraction methods differ; DeepPCT uses GearNet for structure modeling, while MeToken employs its own tokenization approach for encoding micro-environments. Although GearNet was already included as a baseline in our work, DeepPCT’s use of geometric and graph-based structural descriptors is unique to its interaction-focused task. DeepPCT also uses ESM-2 for structural embeddings, whereas MeToken integrates structure through learned sequence-structure representations.
> > >
> > > **WPTCMN/PTCMN [5] Similar to DeepPCT, WPTCMN/PTCMN also focuses on PTM crosstalk prediction rather than single-site PTM prediction.** The modeling of interactions between multiple PTM sites is an important but distinct area of research compared to the residue-specific PTM prediction task addressed by MeToken. While we do not directly compare MeToken to these methods due to the difference in tasks, we will acknowledge their contributions in our revised discussion.
> > >
> > > Ertelt et al. [6] integrates PTM prediction with protein design, aiming to maximize or minimize the likelihood of specific PTMs through structure-based engineering. While this is an important and complementary direction, MeToken is focused on PTM type prediction rather than design optimization. Additionally, Ertelt et al. target specific PTMs and their probabilities, whereas MeToken aims to generalize across diverse modification types.
> > >
> > > PhosHSGN [7] is designed specifically for phosphorylation site prediction, combining sequence and structural information. While PhosHSGN provides valuable insights into phosphorylation-specific models, its scope is narrower than MeToken’s, which is capable of predicting multiple PTM types.

---

> > > > ### Author Response · Authors · 2024-11-22
> > > >
> > > > > [1] Chen, Jie, Ting-Bo Chen, Chen-Qiu Zhang, Ling-Yan Gu, Jia Wang, Yao-Xing Wu, and Jian-Qiang Li. "Capsulated Graph Neural Network for Ubiquitylation Sites Prediction." In 2022 IEEE International Conference on Bioinformatics and Biomedicine (BIBM), pp. 3797-3799. IEEE, 2022.
> > > >
> > > > > [2] Yan, Yu, Jyun-Yu Jiang, Mingzhou Fu, Ding Wang, Alexander R. Pelletier, Dibakar Sigdel, Dominic CM Ng, Wei Wang, and Peipei Ping. "MIND-S is a deep-learning prediction model for elucidating protein post-translational modifications in human diseases." Cell reports methods 3, no. 3 (2023).
> > > >
> > > > > [3] Yu-Xiang Huang, Rong Liu, Improved prediction of post-translational modification crosstalk within proteins using DeepPCT, Bioinformatics, 2024.
> > > >
> > > > > [4] Yan, Yu, Dean Wang, Ruiqi Xin, Raine A. Soriano, Dominic CM Ng, Wei Wang, and Peipei Ping. "Protocol for the prediction, interpretation, and mutation evaluation of post-translational modification using MIND-S." Star Protocols 4, no. 4 (2023): 102682.
> > > >
> > > > > [5] Dai, Yuhao, Lei Deng, and Fei Zhu. "A model for predicting post-translational modification cross-talk based on the Multilayer Network." Expert Systems with Applications 255 (2024): 124770.
> > > >
> > > > > [6] Ertelt, Moritz, Vikram Khipple Mulligan, Jack B. Maguire, Sergey Lyskov, Rocco Moretti, Torben Schiffner, Jens Meiler, and Clara T. Schoeder. "Combining machine learning with structure-based protein design to predict and engineer post-translational modifications of proteins." PLOS Computational Biology 20, no. 3 (2024): e1011939.
> > > >
> > > > > [7] J. Lu, H. Chen and J. Qiu, "PhosHSGN: Deep Neural Networks Combining Sequence and Protein Spatial Information to Improve Protein Phosphorylation Site Prediction," in IEEE Access, vol. 12, pp. 100611-100627, 2024.
> > > >
> > > > We sincerely thank you for your detailed suggestions and for sharing these highly relevant references. Your feedback has greatly enriched our understanding of recent advancements in the field and will help us position MeToken more effectively within the PTM prediction landscape. We will incorporate these discussions into the revised manuscript to ensure a balanced and thorough literature review.
> > > >
> > > > If you have any additional suggestions or feedback, we would be happy to address them. Thank you again for your constructive guidance.
> > > >
> > > > Best regards,
> > > >
> > > > The Authors

---

> ### Author Response · Authors · 2024-11-21
>
> Dear Reviewer GBsm,
> Your response is very meaningful and has helped make our work more convincing.
> Given that the discussion deadline is approaching, we hope that our additional experiments and explanations have addressed your concerns and improved our work. If you are satisfied with our additions, we sincerely hope you will consider increasing our score. If you believe there are still issues with our additions, we kindly ask you to engage in further discussion with us.
> We once again sincerely thank you for your review work, which is of great significance.

---

### Official Review · Reviewer_HxpX · 2024-11-02

**Soundness:** 2
**Presentation:** 2
**Contribution:** 2
**Rating:** 6
**Confidence:** 5

**Summary:**

This paper compiles a large-scale sequence-structure PTM dataset and also propose METoken which uses sub-codebooks for each of the subtypes and temperature scaled vector quantization. On top of the the codebooks and learned tokens for micro-environment of the protein positions it then predicts PTM and its type.

**Strengths:**

The ablation and comparisons are done and results are reported on various useful metrics like F1, MCC and others.

**Weaknesses:**

It is not clear how the sequence similarity or redundancy was mitigated in the train-test-validation sets. Please report how minimal redundancy was ensured among test-train and validation sets.

In the rare modification class, how good is the predictor to discriminate the classes is not made clear. In this case authors may consider to report the different metrics for the PTM classes merged.

There are datasets and methods which are specifically learned for particular type of PTM and that has been a trend in the literature. Please report how does those methods perform comapred to METoken.

**Questions:**

How is the imbalance in the samples for no modification class affects the classification? Particularly the merged small PTMs and no modifications.

---

> ### Author Response · Authors · 2024-11-17
> **Response to Reviewer HxpX (1/2)**
>
> Dear Reviewer HxpX,
>
> Thank you for your feedback and constructive suggestions. We appreciate the opportunity to clarify and expand on several aspects of our work. Below, we address each of your points in detail and outline the changes we plan to make in the revised manuscript:
>
> **Q1** It is not clear how the sequence similarity or redundancy was mitigated in the train-test-validation sets. Please report how minimal redundancy was ensured among test-train and validation sets.
>
> **A1** As shown in **Figure 7 (Appendix C)**, we applied MMseqs2 with a 40% sequence identity threshold, meaning that sequences sharing more than 40% similarity were assigned to the same cluster. After clustering, we assigned entire clusters to either the training, validation, or test set, ensuring that no sequence in one split had more than 40% similarity to any sequence in another split.
>
> **Q2** In the rare modification class, how good is the predictor to discriminate the classes is not made clear. In this case authors may consider to report the different metrics for the PTM classes merged.
>
> **A2** Thank you for your questions. We recognize the importance of clearly demonstrating the model’s ability to handle rare PTM classes and appreciate the opportunity to address this. Rare PTM types pose a significant challenge due to their extreme data sparsity, which makes it difficult for the model to learn meaningful, class-specific features. In the original implementation, we merged all rare PTM types (those with fewer than 100 samples) into a single "rare classes" category to mitigate issues caused by insufficient data per individual class.
>
> In response to your suggestion, we conducted additional experiments where we retained individual PTM types as separate classes rather than merging rare PTMs. The results are summarized below:
>
> | Method | F1     |  Accuracy | Precision | Recall | MCC    | AUROC  | AUPRC  |
> | -------------------------------- | -------- | --------- | ------ | ------ | ------ | ------ | ------ |
> | MeToken | 0.5040 | 0.8991   | 0.5808    | 0.4793 | 0.7603 | 0.9220 | 0.5126 |
> | MeToken w/ all rare PTMs | 0.3045 | 0.9015   | 0.3657    | 0.3031 |  0.7666 | 0.5745 | 0.2335 |
>
> When rare PTMs are treated as distinct classes, the total number of classes increases from 26 to 73, introducing significant class imbalance and making it harder for the model to correctly classify rare PTMs.
>
> **Q3** There are datasets and methods which are specifically learned for particular type of PTM and that has been a trend in the literature. Please report how does those methods perform comapred to MeToken.
>
> **A3** We summarize the characteristics of our dataset and existing PTM datasets in **Table 6 (Appendix C)** of the manuscript. Our dataset is one of the most comprehensive resources available for PTM prediction, containing 1,263,935 PTM sites across 187,812 proteins, spanning 72 PTM types. This not only covers a broader range of modification types than most existing datasets but also integrates both sequence and structural information, enabling models to capture both sequence motifs and spatial context. For comparison:
> - PTMint contains 2,477 PTM sites from 1,169 proteins and covers only six PTM types, which are subsets of those included in our dataset. While it includes structural information, the dataset size limits its applicability for generalized PTM prediction.
> - qPTM provides 660,030 PTM sites across 40,728 proteins, also limited to six PTM types, but lacks structural annotations, restricting its use for models incorporating 3D context.
> - PRISMOID and PTM Structure datasets are small-scale datasets containing structural annotations but cover fewer modification types (37 and 21, respectively) and a limited number of PTM sites and proteins.
> - dbPTM is a very large-scale sequence-only dataset (2,235,000 PTM sites, 223,678 proteins) spanning 72 PTM types. However, it lacks structural annotations, which are crucial for capturing the 3D spatial context of PTMs.
>
> | **Dataset**      | **Number of Sites** | **Number of Proteins** | **Modification Types** | **Sequence** | **Structure** |
> |-------------------|---------------------|-------------------------|-------------------------|--------------|---------------|
> | **PTMint**       | 2,477               | 1,169                   | 6                       | ✔️           | ✔️             |
> | **PTM Structure** | 11,677             | 3,828                   | 21                      | ✔️           | ✔️             |
> | **PRISMOID**      | 17,145             | 3,919                   | 37                      | ✔️           | ✔️             |
> | **qPTM**          | 660,030            | 40,728                  | 6                       | ✔️           | /             |
> | **dbPTM**         | 2,235,000          | 223,678                 | 72                      | ✔️           | /             |
> | **Ours**          | 1,263,935      | 187,812            | 72                  | ✔️           | ✔️             |

---

> ### Author Response · Authors · 2024-11-17
> **Response to Reviewer HxpX (2/2)**
>
> Our dataset is therefore uniquely positioned to serve as a foundation for PTM prediction models that incorporate both sequence and structure, with a focus on addressing the long-tail distribution of PTM types.
>
> In our baseline comparisons, we evaluated MeToken against several state-of-the-art PTM prediction models. Musite-Deep, DeepTL-Ubi, and DeepGpgs are all binary classification models specifically designed to predict a single type of PTM—phosphorylation, ubiquitylation, and glycosylation, respectively. On the other hand, models like EMBER and DeepPhos provide more flexibility, as they can predict multiple kinase-specific phosphorylation types. We re-implement these baselines in our multiclass prediction task, and the results are reported in our manuscript.
>
> **Q4** How is the imbalance in the samples for no modification class affects the classification? Particularly the merged small PTMs and no modifications.
>
> **A4** Thank you for raising this important question about how the imbalance in "no modification" samples and rare PTM classes affects classification performance. We conducted additional experiments to analyze the impact of merging rare PTM classes and including "no modification" sites on model performance. Below, we present a comparison among three configurations: (1) the current MeToken model (which merges rare PTM classes and excludes "no modification" sites), (2) a model that does not merge rare PTM classes, and (3) a model that considers "no modification" sites in addition to PTM classes. The results are summarized below:
>
> |                              | F1     |  Accuracy | Precision | Recall | MCC    | AUROC  | AUPRC  |
> | ---------------------------- | -------- | --------- | ------ | ------ | ------ | ------ | ------ |
> | MeToken                      | 0.5040 | 0.8991   | 0.5808    | 0.4793 | 0.7603 | 0.9220 | 0.5126 |
> | w/o merging rare classes     |  0.3045 | 0.9015   | 0.3657    | 0.3031 | 0.7666 | 0.5745 | 0.2335 |
> | w/ no-modification |  0.1406 | 0.9684   | 0.1383    | 0.1429 | 0.0000 | 0.1752 | 0.0443 |
>
> In our dataset, there are 106,162,623 "no modification" sites compared to 1,263,935 PTM sites, meaning "no modification" accounts for over 98.8% of the data. This extreme imbalance severely impacts model performance.
>
> ---
>
> Your initial review and encouraging feedback were impressive, and we sincerely appreciate your thoughtful and thorough evaluation of our work. We respectfully inquire if there might be an opportunity for you to consider further raising the score? Your kindness and fairness in assessing our manuscript have been invaluable to us.

---

> > ### Comment · Reviewer_HxpX · 2024-11-28
> >
> > Now, is there any way out from to solve this imbalance, like to learn about a context / window that might give a proper justification to the problem? Alternatively, what if synthetic data is used? practically with that low f1, do you suggest the tool to be used ?

---

> ### Author Response · Authors · 2024-11-21
>
> Dear Reviewer HxpX,
> Your response is very meaningful, as it has helped us further clarify certain points in the paper.
> The discussion deadline is approaching, and we kindly inquire whether our explanation has effectively clarified our paper. If our explanation satisfies you, we would appreciate it if you could consider increasing our score. If not, we welcome further discussion with you.
> Thanks for your reviewing work again.

---

> ### Author Response · Authors · 2024-11-24
>
> Dear Reviewer HxpX,
>
> Thanks again for your recognition of our work. Since the deadline for the discussion period draws near, we respectfully inquire whether our response has addressed your concerns. If our response echoes your concerns and issues, we would be grateful if you could consider raising our work's scores?
>
> We sincerely appreciate your contribution and work as a reviewer again.
>
> Sincerely,
>
> The Authors

---

### Official Review · Reviewer_fwnN · 2024-11-03

**Soundness:** 2
**Presentation:** 3
**Contribution:** 2
**Rating:** 5
**Confidence:** 4

**Summary:**

MeToken introduces a novel approach to PTM prediction by integrating sequence and structural data through micro-environment tokenization and a uniform codebook, enhancing model performance across diverse PTM types. The model shows good results but relies on AlphaFold data and lacks evaluation on biological interactions, suggesting areas for future refinement.

**Strengths:**

1. Proposes MeToken that effectively integrates sequence and structural data in PTM prediction. This foresighted approach to using micro-environment tokenization and uniform sub-codebook construction demonstrates promising results in addressing the long-tail distribution of PTM types.
2. Extensive experiments across datasets validate the model's performance, showing significant improvements in metrics such as F1-score and AUPRC over competitive baselines. The experiments reveal that MeToken captures complex biochemical contexts well, potentially setting a new benchmark for PTM prediction.

**Weaknesses:**

1. The reliance on AlphaFold-generated data may limit its generalizability in real-world applications where experimental structures are less available or differ significantly.
2. Although the authors address the dimensionality reduction challenge, computational complexity remains an issue, especially given the additional steps in codebook learning and temperature-scaled quantization. Further exploration of optimization methods would enhance applicability for high-throughput or large-scale data analysis.
3. The study focuses primarily on PTM types without including interactions with enzymes or other proteins, which are crucial for PTM formation and regulation in vivo. Extending the approach to consider these interactions could improve the model’s utility in biological applications.

**Questions:**

MeToken's design, with its uniform sub-codebooks and temperature-scaled quantization, may introduce significant computational demands, especially in high-dimensional cases. Have you explored options like pruning or adaptive quantization to mitigate these costs? This could clarify MeToken’s efficiency for large-scale deployment. Another question is that as protein-protein interactions, particularly with enzymes often influence PTM sites, could you consider ways to include these relationships in MeToken’s predictions? Addressing this would potentially expand the model’s real-world applicability and accuracy in cellular contexts.

---

> ### Author Response · Authors · 2024-11-17
> **Response to Reviewer fwnN**
>
> Dear fwnN,
>
> Thank you for your detailed and insightful feedback. We appreciate your recognition of the strengths of our approach, and your constructive suggestions highlight important areas for further improvement. Below, we address each of your comments and questions:
>
> **Q1** The reliance on AlphaFold-generated data may limit its generalizability in real-world applications where experimental structures are less available or differ significantly.
>
> **A1** Thank you for raising this important concern about the reliance on AlphaFold-generated data. As shown in **Appendix C**, we prioritized experimental data from the PDB whenever it was available. Specifically, we used experimental structures as the primary source of structural information to ensure the highest level of accuracy and biological relevance in our dataset.
>
> We acknowledge, however, that AlphaFold predictions are generated under certain assumptions and are not always perfect representations of in vivo protein structures, especially for highly dynamic regions or protein-protein interfaces. To assess the impact of using AF2 data, we conducted additional evaluations (**Appendix F.3**) to compare the predictive performance on subsets of the dataset containing only experimental structures versus those with AlphaFold2-predicted structures. Our analysis showed that the inclusion of AF2 predictions substantially improved the size and diversity of the dataset, enabling more robust training
>
> **Q2** Although the authors address the dimensionality reduction challenge, computational complexity remains an issue, especially given the additional steps in codebook learning and temperature-scaled quantization. Further exploration of optimization methods would enhance applicability for high-throughput or large-scale data analysis.
>
> **A2** Thank you for raising this important point. We will include a discussion of these limitations, along with the potential optimization strategies mentioned, in the revised manuscript. Additionally, we are exploring the possibility of open-sourcing a lightweight variant of MeToken that trades off some accuracy for faster runtime, making it suitable for high-throughput or resource-constrained scenarios.
>
> **Q3** The study focuses primarily on PTM types without including interactions with enzymes or other proteins, which are crucial for PTM formation and regulation in vivo. Extending the approach to consider these interactions could improve the model’s utility in biological applications. Could you consider ways to include these relationships in MeToken’s predictions?
>
> **A3** This is an excellent suggestion, and we appreciate the opportunity to expand on this important aspect. You are correct that PTM formation and regulation are often mediated by enzymes (e.g., kinases, glycosyltransferases, proteases) or occur in the context of protein-protein interactions. Incorporating these interactions into MeToken could significantly enhance its biological utility, as the structural micro-environment alone does not fully capture the complex interplay of factors influencing PTMs in vivo.
>
> One potential way to address this is by integrating enzyme recognition motifs or interaction features directly into MeToken’s inputs, such as:
> - We could add known enzyme-specific sequence motifs (e.g., kinase binding motifs for phosphorylation) as auxiliary features during training.
> - Including protein-protein interaction network data could provide additional context, as PTMs often occur at protein interfaces or are influenced by specific interaction partners.
> - Utilizing reaction-specific enzyme pockets generation approaches for enzymes and substrates could augment the micro-environment representation.
>
> **Q4** Have you explored options like pruning or adaptive quantization to mitigate these costs?
>
> **A4** Thank you for this insightful suggestion. Pruning and adaptive quantization are promising strategies to reduce the computational costs associated with codebook learning and tokenization. While these methods were not explored in the current study, they could significantly improve MeToken’s efficiency without substantial loss of accuracy. We appreciate your thoughtful questions and suggestions, which will help us significantly improve the clarity and scope of the manuscript.
>
> ----
>
> We express our sincere gratitude for your constructive feedback in the initial review. It is our hope that our responses adequately address your concerns. Your expert insights are invaluable to us in our pursuit of elevating the quality of our work. If you find our efforts satisfactory, we kindly request that you consider revisiting and potentially raising the score for our submission. Thank you once again for your time and valuable input.

---

> ### Author Response · Authors · 2024-11-21
>
> Dear Reviewer fwnN,
> Your response is helpful and be appreciated for us, making us refine and clarify some points in our work.
> As the time passing by, the deadline of discussion period is coming soon. If our response addresses your concerns, we hope you will consider increasing the score of our work. We appreciate your time for this.
> We would like to express our gratitude once again for your reviewing work.

---

> ### Author Response · Authors · 2024-11-24
>
> Dear Reviewer fwnN,
>
> Thank you once again for your review. As the discussion period deadline approaches, we are eager to hear your opinions and suggestions regarding our response. If our reply addresses your concerns, we would greatly appreciate it if you would consider raising the score of our work?
>
> Your recognition and feedback are extremely valuable to us.
>
> Sincerely,
>
> The Authors

---

### Official Review · Reviewer_sns6 · 2024-11-04

**Soundness:** 3
**Presentation:** 3
**Contribution:** 3
**Rating:** 5
**Confidence:** 3

**Summary:**

This paper proposed a pipeline to predict PTMs (mostly in residue level as opposed to other global level) by addressing the issue of lacking structural information in the modeling. Additionally they curated a new dataset for such task as added contribution. Through comparison with baselines and some ablation studies, they show the better performance of the method

**Strengths:**

The raised challenges in the current landscape of PTMs prediction are legit. These issues make both modeling and evaluation in the domain difficult, thus the claim for such challenge is of high relevance.

The introduction of MeToken Model, that tokenizes amino acids’ micro-environments by integrating both sequence and structural information into discrete tokens, is relevant (although it seems like most of the boost is from this innovation so it begs the questions why the latter two, see below Weakness comments)

To handle the long-tail distribution of PTM types, MeToken introduces a sub-codebook strategy which arguably serves as a regularizer for the model to learn canonical representations

**Weaknesses:**

The hypothesis that incorporating structural data into PTM prediction can improve accuracy is well-supported by prior research indicating that sequence-only approaches may overlook important spatial context. But from table 4 it seems like this has enough boost compared to other baselines. It's beneficial to investigate deeper (and perhaps more ablation studies on only this arm of contribution) to stress the importance of added benefit of structural information.

Additionally, ESM-2 baseline seems to be very close the proposed method and beats all other baselines so there's still arguably a lot of sequence information that's useful already, so how to showcase their method is not overfitting to structural features might help reviewers to understand the validity of their method.

Again, the latter contributions, uniform sub-codebook strategy and temperature-scaled vector quantization, seems to offer marginal benefit which IMO dilutes the core contribution brought by the Micro-env arm of contribution. In addition, the temperature adjusted approach might necessitate a huge effort for hyperparameter tunning to make it work. But the authors didn't discuss any ramification of it. Further, the uniform sub-codebook strategy might risk embedding redundancy for common PTM types, and this was not discussed either.

**Questions:**

They highlighted the long tail nature of the current datasets in the domain but it lacks some deeper analysis for the distribution e.g. rarity analysis, frequency statistics. It's unclear the extent to which the long tail nature is currently at play for the modeling difficulty in the domain, as what they claimed as a challenge. Additionally, reducing the class imbalance by consolidating PTM types with fewer than 100 samples into a single “rare sites” category might be exacerbate the imbalance issue? The rare classes are not modeled/studied individually then

Also, UMAP is often better at preserving global structure in the final projection. This means that the inter-cluster relations are potentially more meaningful than in t-SNE. So might want to suggest the authors to compare that visualization (see appendix c: https://arxiv.org/pdf/1802.03426)

---

> ### Author Response · Authors · 2024-11-17
> **Response to Reviewer sns6 (1/3)**
>
> Dear sns6,
>
> Thank you for your detailed and constructive feedback on our work. We appreciate your insights into both the strengths and weaknesses of our approach, and we address each of your points below:
>
> **Q1** It's beneficial to investigate deeper (and perhaps more ablation studies on only this arm of contribution) to stress the importance of added benefit of structural information.
>
> **A1** Thank you for raising this important point. To better illustrate the contribution of structural information in our model, we performed additional ablation experiments where we removed the structural information from MeToken and evaluated the performance. The results are summarized below:
>
> | Method | F1     | Accuracy | Precision | Recall |  MCC    | AUROC  | AUPRC  |
> | --------------- | -------- | --------- | ------ | ------ | ------ | ------ | ------ |
> | MeToken         | 0.5040 | 0.8991   | 0.5808    | 0.4793 | 0.7603 | 0.9220 | 0.5126 |
> | MeToken_w/o_str | 0.1956 | 0.8745   | 0.1894    | 0.2300 |  0.7132 | 0.8953 | 0.2170 |
>
> These results clearly demonstrate the significant added benefit of incorporating structural information into MeToken.
>
> **Q2** Additionally, ESM-2 baseline seems to be very close the proposed method and beats all other baselines so there's still arguably a lot of sequence information that's useful already, so how to showcase their method is not overfitting to structural features might help reviewers to understand the validity of their method.
>
> **A2** Thank you for this thoughtful observation. We agree that ESM-2, as a powerful sequence-based model, does indeed provide valuable insights based on sequence information alone. However, it is important to highlight several key distinctions between our method and ESM-2. ESM-2-650M, which we used as a baseline, contains over 650 million parameters and has been pretrained on 65 million protein sequences. In contrast, our model is lightweight, with fewer than 10 million parameters, and it is trained on a dataset of approximately 140k data. Despite these differences, our method achieves competitive performance with ESM-2 and even outperforms it on certain structure-dependent tasks, demonstrating that structural information adds meaningful context to PTM prediction.
>
> To showcase the added benefit of incorporating structural information, we examined the tokens within our learned codebook, particularly focusing on those that represent amino acid micro-environments. As illustrated in **Figure 20 (in Appendix F.4)** of our manuscript, the most prevalent amino acids associated with a specific token are valine (Val, V) and leucine (Leu, L), both of which are hydrophobic and share similar structural properties. The distribution also includes three other hydrophobic amino acids, reinforcing this trend. This pattern suggests that our model has successfully learned not only the sequence-specific characteristics of these amino acids but also their associated structural properties, such as their tendency to cluster in hydrophobic regions of proteins.
>
> By explicitly capturing these structural and physicochemical features in the form of learned tokens, our method introduces a level of structural awareness that goes beyond simple sequence-based representations. The structure of the token space reflects the functional and structural context of the amino acid environments, which provides a richer understanding of the micro-environment around PTM sites.

---

> ### Author Response · Authors · 2024-11-17
> **Response to Reviewer sns6 (2/3)**
>
> **Q3** Uniform sub-codebook strategy and temperature-scaled vector quantization, seems to offer marginal benefit which IMO dilutes the core contribution brought by the Micro-env arm of contribution. In addition, the temperature adjusted approach might necessitate a huge effort for hyperparameter tunning to make it work. But the authors didn't discuss any ramification of it. Further, the uniform sub-codebook strategy might risk embedding redundancy for common PTM types, and this was not discussed either.
>
> **A3** Thank you for raising these important points. We appreciate your careful consideration of the uniform sub-codebook strategy and temperature-scaled vector quantization. We understand that these techniques may seem to offer only marginal benefits compared to the core contribution of the micro-environment tokenization. However, we believe that they play an important role in improving the overall effectiveness of the model and in refining the functional token space for PTM prediction.
>
> The uniform sub-codebook strategy was designed to address the challenges posed by the long-tail distribution of PTM classes. By dividing the space into smaller, more manageable sub-codebooks, we were able to prevent the model from overfitting to highly frequent classes while still allowing it to learn generalizable features. While we acknowledge that this strategy could potentially introduce redundancy for common PTM types, it is important to note that this redundancy is mitigated by the way the sub-codebooks are structured. Each sub-codebook is tailored to capture specific structural or functional features relevant to the PTM classes it represents. The redundancy issue you raised is something we will explore further in future work to ensure the sub-codebook strategy delivers its intended benefits without unnecessary overlap. **We have added this discussion in the limitation section of our revised manuscript.**
>
> We are sorry that the detailed temperature-adjusting implementation is missing. In our implementation, the initial temperature is set to 1 and then linearly decays to 0.0001 over the epochs of training.
>
> We fully recognize that these strategies may appear as incremental or potentially adding complexity, and we appreciate your feedback on this. In future work, we will investigate additional methods to refine these techniques and better assess their true impact on model performance. We really appreciate your insights into their potential limitations.
>
> **Q4** They highlighted the long tail nature of the current datasets in the domain but it lacks some deeper analysis for the distribution e.g. rarity analysis, frequency statistics. It's unclear the extent to which the long tail nature is currently at play for the modeling difficulty in the domain, as what they claimed as a challenge.
>
> **A4** Thank you for this insightful question regarding the long-tail distribution of PTM classes and its impact on modeling difficulty.
>
> As shown in **the top half of Figure 8 (Appendix C)** of the manuscript, the dataset exhibits a pronounced long-tail distribution. The most frequent PTM class, *Phosphorylation*, has a scale of over $10^6$ labeled sites, making it by far the dominant class in the dataset. In contrast, the rarest classes, such as *Carboxyethylation, D-glucuronoylation, N-carbamoylation, Octanoylation, Pyrrolylation, and S-carbamoylation*, have fewer than $10^1$ labeled instances each. These classes represent extreme sparsity, with fewer than 10 samples per modification type, making it almost impossible for a model to learn reliable representations for these rare classes without additional interventions.
>
> To provide more clarity, we summarize the distribution of the top five most frequent PTM types in the dataset below (total=**1,263,935** sites):
>
> | Modification                       | Count  |
> | --- |---|
> | Phosphorylation                    | 969,887 |
> | Ubiquitination                     | 141,809 |
> | Acetylation                        | 92,684  |
> | N-linked Glycosylation             | 12,704  |
> | Methylation                        | 9,149   |
>
> - Phosphorylation constitutes **76.74%** of the dataset.
> - Classes like Acetylation and Ubiquitination follow with significant numbers, but they are still orders of magnitude smaller than Phosphorylation.
> - Rare PTMs, as noted above, all of them collectively contribute less than 0.08% of the dataset.

---

> > ### Author Response · Authors · 2024-11-17
> > **Response to Reviewer sns6 (3/3)**
> >
> > **Q5** Reducing the class imbalance by consolidating PTM types with fewer than 100 samples into a single “rare sites” category might be exacerbate the imbalance issue?
> >
> > **A5** The long-tail distribution in PTM datasets is a significant challenge, as the rarest PTM types often have extremely limited data. Many of these classes contain fewer than 20 instances, making it difficult for the model to learn meaningful class-specific features. In our dataset, **all rare PTM classes combined represent only about 1,000 sites out of a total of 1,263,935 sites, accounting for approximately 0.08% of the dataset.** By grouping these rare PTMs into a single "rare sites" category, we increase the effective representation of this group and ensure that the model has sufficient data to learn generalized features that capture patterns shared across rare PTMs.
> >
> > This grouping is not merely a form of data aggregation; based on our visualization result, we make a hypothesis that many rare PTMs, despite their distinct biochemical and biological roles, might share underlying structural or functional properties. We are here to make some assumptions, For example:
> > - Some rare PTMs might be recognized by similar enzymatic machinery.
> > - Others might occur in comparable structural environments, such as requiring specific residue accessibility or unique structural motifs.
> >
> > In Figure 6 of our manuscript, we show that **rare PTMs cluster together in the model’s codebook latent space, might be indicating that the model is learning shared structural and functional patterns to some extent**. This works as a possible assumption here, supports our hypothesis that consolidating rare PTMs may help the model capture and keep useful generalizations and shared features that are otherwise lost when training on individual rare PTM types with very limited data.
> >
> > We acknowledge the trade-offs involved and are committed to exploring more sophisticated methods in future research to further address the challenges of rare PTM prediction.
> >
> > **Q6** UMAP is often better at preserving global structure in the final projection.
> >
> > **A6**  Thank you for this suggestion. We agree that UMAP may provide a clearer representation of global relationships between PTM classes. We have added the UMAP visualization in the **Appendix H** of our revised manuscript.
> >
> > ___
> >
> > We express our sincere gratitude for your valuable feedback during the initial review, especially for your careful attention to detail. Thank you for providing us an opportunity to clarify the points you raised. We are hopeful that our responses adequately address your concerns. The insights you provide are indispensable to us, enabling significant improvements in the quality of our work. We would be deeply grateful if you could consider raising the score.

---

> ### Author Response · Authors · 2024-11-21
>
> Dear Reviewer sns6,
> We appreciate your response, which has helped us further improve our work. Your time and contribution means a lot for us.
> At the same time, if our explanation has addressed your concerns, we kindly hope that you would consider increasing the score or confidence of our work. If you still have any questions regarding our work, please feel free to contact us, and we will respond as soon as possible.
> Thanks again for your assistance here.

---

> ### Comment · Reviewer_sns6 · 2024-11-22
>
> I thank authors for addressing my points. But my main concerns still stands (1) core contribution is diluted, which should've been a heavy study on WHY the structural data works better, as this finding can drive more interesting research, rather than merely cranking up the prediction performance by incorporating the added contribution (the latter two) (2) the smaller model size is another interesting angle but it feels weak to say 'our model is much smaller and at the same time offers few points bump'. In the foundation model era, scaling up the model size to have better generalizability is more important (unless you are in some edge computing domains). Plus the model parameters wasn't a driving factor for contribution in the paper to begin with so mentioning it wouldn't help reviewers to appreciate the core technical novelty. Appendix F.4 offers a tiny snapshot, which covers a limited number of amino acids, thus less convincing

---

> > ### Author Response · Authors · 2024-11-22
> >
> > Thank you for your thoughtful follow-up and for raising these important concerns. We greatly appreciate your feedback and the opportunity to further clarify our contributions and address the points you’ve highlighted.
> >
> > 1. The Role of Structural Data
> >
> > We understand and agree with your point that a deeper exploration of why structural data improves PTM prediction would significantly enhance the impact of our study. While structural data is commonly leveraged in tasks such as protein function prediction and protein-protein interaction modeling, it is underexplored in the specific context of PTM prediction, which often focuses heavily on sequence motifs alone. To address this gap, we introduced the micro-environment tokenization approach, which integrates both sequence and structural information into discrete tokens. This unified tokenization enables the model to encode residue-level context in a way that incorporates spatial relationships, sequence motifs, and local environmental properties simultaneously.
> >
> > You also raised a valid point about how the "latter two contributions"—the uniform sub-codebook strategy and temperature-scaled vector quantization—tie into this broader role of structural data. **We would like to clarify that these two components were designed as complementary mechanisms to enhance the encoding of the sequence-structural context into the discrete token representations. They are not standalone contributions but rather integral parts of the tokenization suite. Together, they ensure that rare PTMs are adequately represented (via the sub-codebook strategy) and that the quantization process remains flexible and efficient during training (via temperature-scaled vector quantization).** These components enhance the model’s ability to capture structural subtleties and generalize effectively across diverse PTM types.
> >
> > 2. Model Size and Its Role in the Contribution
> >
> > We appreciate your comments regarding the relevance of model size in the current "foundation model" era, where larger models are often prioritized for better generalizability. We fully understand that scaling up models can drive broader applicability, but we believe that the smaller model size still plays a meaningful role in the specific context of this study.
> >
> > That said, we believe the smaller model size remains relevant in the specific context of PTM prediction for the following reasons:
> > * Task-Specific Constraints: PTM prediction is a domain-specific task where the input size (e.g., sequence and structural context of residues) is inherently constrained by biological factors. Unlike general protein modeling tasks (e.g., structure prediction or sequence design), PTM prediction focuses on residue-level classification. In this scenario, the marginal gains from scaling up model size may not justify the increased computational cost, especially when a smaller, task-focused model is sufficient to achieve state-of-the-art performance.
> > * Empirical Validation Across Datasets: **The ability of our smaller model to generalize effectively has been demonstrated by training on our large-scale dataset and directly evaluating on external datasets like PTMint and qPTM, which differ significantly in data distribution and experimental conditions.** Despite the differences, MeToken achieved strong performance without requiring larger model architectures. This result highlights the model’s ability to leverage rich sequence-structure data effectively while maintaining computational efficiency.
> > * Efficiency and Practical Applicability: 	While large models are becoming the standard in many protein modeling tasks, they are not always ideal for domain-specific applications, particularly when computational resources are a concern. For example, PTM prediction is often integrated into larger workflows, such as proteomics pipelines or drug discovery platforms, where efficiency matters. In these scenarios, a smaller, high-performing model like MeToken can offer practical advantages.
> >
> > Thank you once again for your thoughtful feedback and for giving us the opportunity to clarify these points. We believe that our work demonstrates the value of structural information in PTM prediction and highlights how a carefully designed, smaller model can achieve competitive results in a domain-specific context.
> >
> > Sincerely,
> >
> > The Authors

---

### Official Review · Reviewer_nhjP · 2024-11-04

**Soundness:** 4
**Presentation:** 3
**Contribution:** 4
**Rating:** 8
**Confidence:** 4

**Summary:**

The work aims at creating a structure and sequence-based representation of microenvironment  of the  residues and use this for post-translational modification (PTM) classification. They compile a large-scale sequence-structure PTM dataset featuring over 1.2 million residue-level annotated sites across multiple PTM types. The method MeToken tries to capture the context of the site using the sequential and structural neighbourhood of the modified residue. The paper learns a codebook that represents the high-dimensional data into simpler, representative tokens. The work is interesting contribution to the field.

**Strengths:**

- The authors use valid strategies for experimental evaluation. For example, they take into account sequence similarity to prevent leakage in the evaluation. Uses appropriately macro-avearged evaluation metrics. Many decisions are based on valid reasons or experimentation (using ESM-2 vs ESM3) and (PDB vs AlphaFold+PDB vs AlphadFold)
- Results on two external datasets are provided.
- The work is interesting in the sense that it uses the structural features of the modified site together with the sequence
- Authors compare against existing PTM models and wider protein language models. They have modified some of the specialized models such as DeepPhos to evaluate for the general PTM classification task.
- Authors provide an ablation study to understand the contribution of the different parts of the method

**Weaknesses:**

Weakness:

A simple baseline of one-hot encoded amino acid sequence representations would be useful to see if certain classes can easily be predicted based on their modification site.

The authors could compare their work to other structure-aware representations. For example, SAProt using the Foldseek structural tokens learns a sturture aware embedding.

The authors demonstrate one of the codebooks as an example but a more extensive analysis on this would be interesting. For example a very basic question would be : "Are the amino acid preferences (K for Sumoylation, S/T/Y for Phosphorylation, etc )correctly captured by the codebooks?"

Information on the hyperparameters used to train MeToken and the baseline models are needed for reproducibility of the results.

A hyperparameter sensitivity analysis would be useful. For example the codebook size, the number of neighbors in the k-nn graph etc.

Distinguishing some PTMs from others are easier because of the distinct amino acid preferences in their modification sites. Hard evaluation sets up could have been designed to understand the weakness and the strength of the methodology. For example, does the model correctly differentiate a Lysine residue  being ubiquninated or sumoylated?

The authors do not provide results on performance across different PTM classes. Are certain PTMs are much easier because of the available training data?

**Questions:**

Questions
How are the no-modification sites determined? Are these all sites in protein sequences where no modifications are reported or are they subsets based on amino acid types?

Is the quantization technique and the codebook strategy different than the cited work FoldToken? The differences and similarities should be included in the relevant literature part as it is very closely related to the work introduced here.
Of the many structures available for a protein in the PDB, which one is selected ? What criteria are applied on this selection?
Authors suggest that they introduce a sub-codebook strategy to handle the long-tail distribution but at the end of the day, all the rare classes are grouped as a single class. How does that help?

---

> ### Author Response · Authors · 2024-11-17
> **Response to Reviewer nhjP (1/5)**
>
> Dear Reviewer nhjP,
>
> Thank you for your thoughtful and detailed feedback. We appreciate your positive assessment of the strengths of our work, as well as your suggestions for areas of improvement. We address each of your comments in detail below:
>
> **Q1** A simple baseline of one-hot encoded amino acid sequence representations would be useful to see if certain classes can easily be predicted based on their modification site.
>
> **A1** This is an excellent suggestion. We agree that including a one-hot encoded baseline for amino acid sequences would provide a valuable comparison, helping us to understand the baseline predictability of PTM classes based purely on sequence information without complex embeddings. We have now added a one-hot encoding baseline.
>
> For the one-hot encoding baseline, we implemented a simple model with:
> - An embedding layer that converts the one-hot encoded amino acid sequence into dense vectors,
> - Three fully connected linear layers (hidden dimension=128) with ReLU activation functions in between, and
> - A final classification layer to predict the PTM type.
>
> This baseline model represents a straightforward approach to PTM prediction based solely on primary sequence data. Below, we summarize the results of this one-hot baseline in comparison to MeToken, our proposed model.
>
> | Method |F1     |  Accuracy | Precision | Recall |  MCC    | AUROC  | AUPRC  |
> | --- | -------- | --------- | ------ | ------ | ------ | ------ | ------ |
> | MeToken | 0.5040 | 0.8991   | 0.5808    | 0.4793 | 0.7603 | 0.9220 | 0.5126 |
> | One-hot | 0.1764 | 0.8736   | 0.1644    | 0.2149 | 0.7098 | 0.8893 | 0.2077 |
>
> The results indicate that while the one-hot baseline achieves reasonable accuracy and AUROC scores, it performs significantly worse in terms of precision, recall, and F1 scores. This suggests that one-hot encoding struggles to capture the complex patterns required for effective PTM prediction, particularly for rare or structurally influenced PTM types.
>
> To further explore this baseline’s limitations, we evaluated its performance on specific PTM types. Below are the results for phosphorylation, ubiquitylation, and acetylation, which are among the most biologically relevant PTM types.
>
> | Method  | Accuracy | Precision | Recall | F1     |
> | -----   | -------- | --------- | ------ | ------ |
> | Onehot - phosphorylation | 0.9866   | 0.1667    | 0.1644 | 0.1655 |
> | Onehot - ubiquitylation  | 0.2990   | 0.1667    | 0.0498 | 0.0767 |
> | Onehot - acetylation     | 0.9699   | 0.1667    | 0.1617 | 0.1641 |
>
> These results highlight some important observations:
>
> - Phosphorylation: The one-hot baseline achieves relatively high accuracy because phosphorylation sites often have clear sequence motifs (e.g., serine/threonine/tyrosine residues with flanking sequence preferences). However, its precision and recall are low, reflecting difficulty in distinguishing phosphorylation sites from non-modified residues with similar motifs.
> - Ubiquitylation: The performance on ubiquitylation is poor. This aligns with the biological reality that ubiquitylation does not have clear sequence motifs and is strongly influenced by structural factors, such as residue accessibility and enzymatic binding.
> - Acetylation: Similar to phosphorylation, acetylation achieves decent accuracy but suffers from poor precision and recall, as sequence alone cannot fully capture the modification’s context.
>
> Thank you again for this valuable suggestion, which has helped us better contextualize and strengthen the evaluation of our method.
>
> **Q2** The authors could compare their work to other structure-aware representations. For example, SaProt using the Foldseek structural tokens learns a structure-aware embedding.
>
> **A2** Thank you for suggesting additional comparisons with structure-aware models such as SaProt. We're now conducting experiments comparing MeToken with SaProt’s structure-aware embeddings. The results will be posted here once the experiment done.
>
> **Q3** The authors demonstrate one of the codebooks as an example but a more extensive analysis on this would be interesting. For example a very basic question would be: "Are the amino acid preferences (K for Sumoylation, S/T/Y for Phosphorylation, etc) correctly captured by the codebooks?"
>
> **A3** Thank you for this insightful suggestion. We have expanded our analysis of the codebooks to investigate whether they capture the well-documented amino acid preferences associated with specific PTM types. This analysis highlights how the tokenization process aligns with biological knowledge and how the model handles shared residues across different PTM types.

---

> > ### Comment · Reviewer_nhjP · 2024-11-18
> > **imbalance?**
> >
> > Thank you for your answers. The one-hot encoded results highlight the label imbalance (accuracy, auroc being very high, but aupr being low). That brings the question, is the imbalance properly handled in this one-hot encoded baseline? As some of the difference might be just attributed to not handling it properly.

---

> > > ### Author Response · Authors · 2024-11-18
> > >
> > > Thank you for your follow-up question. This one-hot baseline follows the same experimental setup as other approaches in our manuscript and does not employ any special techniques to specifically address class imbalance. Our intention with the one-hot baseline was to provide a straightforward comparison based purely on sequence information without introducing additional mechanisms that might artificially improve its performance.
> > >
> > > We appreciate your insightful comment and look forward to your further guidance.

---

> > ### Comment · Reviewer_nhjP · 2024-11-27
> >
> > The authors addressed most of my questions (SaProt experiments remains). Thank you. I will increase my score accordingly.

---

> > > ### Author Response · Authors · 2024-11-27
> > >
> > > Dear Reviewer nhjP,
> > >
> > > Thank you for your thoughtful reply and for recognizing our efforts to address your questions. We deeply appreciate your willingness to increase your score—it means a lot to us.
> > >
> > > Regarding the SaProt experiments, we acknowledge that this was an important request, and we appreciate your patience as we worked to complete the analysis. Given the size of our large-scale dataset, the computational demands were substantial, especially during the rebuttal period when resources were heavily utilized. Nevertheless, we conducted the SaProt experiments under the same experimental settings as ESM-2 and ESM-3, as all three models are pretrained protein language models. The results are summarized in the table below:
> > >
> > > |Method|F1|Acc|Pre|Rec|MCC|AUROC|AUPRC|
> > > |-|-|-|-|-|-|-|-|
> > > |ESM2|43.94|89.75|55.67|42.15|75.83|91.52|47.31|
> > > |ESM3|40.65|87.48|45.72|38.67|70.31|89.91|41.36|
> > > |SaProt|45.18|89.87|57.23|43.14|75.66|93.66|48.81|
> > >
> > > These results demonstrate that SaProt achieves slightly better performance than ESM-2 across most metrics, particularly in AUROC and AUPRC, suggesting that incorporating Foldseek’s structure-aware 3D tokens adds value for PTM prediction. However, the improvement is relatively modest, likely due to the limited resolution of Foldseek’s discrete tokens (restricted to only 20 types). This level of granularity may not be sufficient to capture the detailed micro-environmental features required for PTM prediction tasks, where fine-grained structural and biochemical information is critical.
> > >
> > > We have included these findings and their implications in the revised manuscript, providing clarity and additional context for the comparative performance of SaProt.
> > >
> > > **Thank you once again for your constructive feedback and for guiding us toward a more comprehensive evaluation. Your insights have been instrumental in refining our work. We are extremely grateful for your time and effort in reviewing our manuscript!**

---

> ### Author Response · Authors · 2024-11-17
> **Response to Reviewer nhjP (2/5)**
>
> **Codebook and Amino Acid Preferences Analysis**
>
> To analyze whether the codebook effectively captures these preferences, we examined the relationships between amino acids, PTM types, and their token indices. Below, we show the Top 5 groupings based on frequency:
>
> | AA   | Token Index | PTM             | Count |
> | ---- | ----------- | --------------- | ----- |
> | S    | 3197        | Phosphorylation | 65531 |
> | T    | 2393        | Phosphorylation | 23613 |
> | Y    | 2617        | Phosphorylation | 8425  |
> | K    | 1332        | Acetylation     | 6044  |
> | K    | 1403        | Acetylation     | 4425  |
>
> From this table, we observe that the model effectively captures the known amino acid preferences for specific PTMs:
>
> - Phosphorylation predominantly occurs on serine, threonine, and tyrosine residues, which are consistently associated with specific token indices.
> - Acetylation primarily occurs on lysine residues, which are also captured and assigned to specific tokens.
>
> This demonstrates that the tokenization process aligns well with established biochemical knowledge of PTM site preferences.
>
> **Multiple Token Assignments for the Same PTM and Residue**
>
> While the tokenization process captures the general preferences, we also observe that the same modification on the same amino acid can sometimes be mapped to multiple token indices. For example, acetylation on lysine is assigned to five distinct token indices, as shown below:
>
> | AA   | Token Index | PTM         | Count |
> | ---- | ----------- | ----------- | ----- |
> | K    | 1332        | Acetylation | 6044  |
> | K    | 1403        | Acetylation | 4425  |
> | K    | 192         | Acetylation | 870   |
> | K    | 2650        | Acetylation | 532   |
> | K    | 1294        | Acetylation | 35    |
>
> This result reflects the biological complexity of acetylation on lysine, which can occur in diverse micro-environments. Different structural and sequence contexts may result in different token assignments, indicating that the model captures subtleties in the local environments surrounding the modified lysine residues.
>
> **Case Study: Sumoylation on Lysine**
>
> Similarly, we analyzed sumoylation on lysine residues. Sumoylation, like acetylation, also occurs on lysine but involves distinct enzymatic processes. The token assignments for sumoylation on lysine are shown below:
>
> | AA   | Token Index | PTM         | Count |
> | ---- | ----------- | ----------- | ----- |
> | K    | 1332        | Sumoylation | 139   |
> | K    | 1403        | Sumoylation | 32    |
> | K    | 2650        | Sumoylation | 31    |
> | K    | 192         | Sumoylation | 23    |
> | K    | 1294        | Sumoylation | 1     |
>
> Interestingly, some sumoylation sites and acetylation sites are mapped to the same token indices (e.g., Token 1332 and Token 1403). This suggests that the local micro-environments of these lysine residues share structural or sequence similarities. However, as we addressed in Q5, despite these overlaps in tokenization, the downstream predictor is still able to correctly distinguish between these two PTM types, leveraging additional features and learned relationships.
>
> **Q4** Information on the hyperparameters used to train MeToken and the baseline models are needed for reproducibility of the results.
>
> **A4** Thank you for raising this important point. We will include a comprehensive table detailing the specific hyperparameters used for training both MeToken and the baseline models. To summarize briefly, we trained MeToken using the AdamW optimizer with a learning rate of $0.0001$ for a total of 20 epochs. The batch size was fixed at 32. For the codebook, we used a size of $128 \times 26 = 3328$, meaning that each PTM type is assigned a sub-codebook with 128 discrete codes. The initial temperature for the vector quantization process was set to $\tau_v = 1$, and it decayed linearly to $\tau_v = 0.0001$ over epochs.

---

> > ### Comment · Reviewer_nhjP · 2024-11-25
> > **Q3**
> >
> > Authors replied "Interestingly, some sumoylation sites and acetylation sites are mapped to the same token indices (e.g., Token 1332 and Token 1403). " Actually all the token indices are identical. Only the order is different. How do the tokens help differentiate the two types?

---

> > > ### Author Response · Authors · 2024-11-25
> > >
> > > Thank you for your follow-up question and for bringing up this important clarification. We greatly value the opportunity to provide further detail.
> > >
> > > In MeToken, each PTM type is allocated its own sub-codebook consisting of 128 code embeddings. Each sub-codebook lies in the space $\mathbb{R}^{128 \times D}$, where $D$ represents the dimensionality of the code embeddings. These embeddings are learned during training and reflect specific patterns of local micro-environments around residues. **The indices of the embeddings within the sub-codebook act as discrete, compressed representations of these micro-environmental features.**
> > >
> > > Briefly, there is a token embedding behind each token index. These embeddings encode the unique micro-environmental features of each PTM type, allowing the model to effectively differentiate between modifications.

---

> ### Author Response · Authors · 2024-11-17
> **Response to Reviewer nhjP (3/5)**
>
> **Q5** Distinguishing some PTMs from others is easier because of the distinct amino acid preferences in their modification sites. Hard evaluation setups could have been designed to understand the weakness and strength of the methodology. For example, does the model correctly differentiate a lysine residue being ubiquitinated or sumoylated?
>
> **A5** Thank you for pointing this out.
>
> In this experiment, we visualized the final embeddings before prediction (referred to as "prediction embeddings") for lysine residues marked as either ubiquitinated or sumoylated. These embeddings were then analyzed using a SVM classifier to determine how separable these two PTM types are in the learned embedding space. The results showed that the model has a strong ability to distinguish between the two modifications, achieving an accuracy of 0.9881 and an F1-score of 0.99. These metrics indicate a clear separation in the learned embedding space, suggesting that the model has captured the relevant features needed to differentiate ubiquitination from sumoylation. A visualization of this separation is provided in the revised manuscript (**Figure 22 in Appendix I**), offering further clarity.
>
> A similar analysis was conducted for O-glycosylation and phosphorylation, which occur on serine or threonine residues. Despite these modifications sharing similar sequence contexts, the model was again able to distinguish them effectively. The SVM trained on their prediction embeddings achieved an accuracy of 0.9934 and an F1-score of 0.99, further demonstrating the model's capacity to capture nuanced differences in the local micro-environments of residues. These results highlight the ability of the model to differentiate between closely related modification types, even when they occur on the same amino acid residues or share overlapping structural and sequence contexts. **This indicates that the combination of sequence, structural features, and the tokenization process enables the model to learn distinct, generalizable representations.**
>
> **Q6** The authors do not provide results on performance across different PTM classes. Are certain PTMs much easier to predict because of the available training data?
>
> **A6** Thank you for pointing this out. We agree that analyzing the performance of our model across different PTM classes is critical to understanding its strengths and weaknesses. To address this, we conducted additional experiments and evaluated the performance of our model across three major subgroups of PTMs: chemical groups (e.g., phosphorylation, acetylation), proteins/peptides (e.g., ubiquitylation, sumoylation), and complex molecules (e.g., glycosylation).
>
> The results are summarized in the table below:
>
> | Method | F1 Score | Accuracy | Precision | Recall |  MCC Score | AUROC  | AUPRC  |
> | ----------------- | -------- | --------- | ------ | -------- | --------- | ------ | ------ |
> | Chem Groups       | 0.6165 | 0.9533 | 0.6968 | 0.5809 |  0.7640 | 0.9902 | 0.6538 |
> | Protein/Peptide   |  0.3798 | 0.6903 | 0.5579 | 0.3038 | 0.0477 | 0.6851 | 0.4650 |
> | Complex Molecules | 0.3470 | 0.7071 | 0.6667 | 0.3344 |  0.4850 | 0.6654 | 0.6643 |
>
> Certain PTM types are indeed easier to predict due to the availability of training data and clearer sequence motifs. For example, chemical group modifications like phosphorylation benefit from abundant data and well-characterized motifs, leading to higher predictive performance. PTMs with weaker sequence motifs or complex structural dependencies (e.g., protein/peptide and complex molecule modifications) are more challenging for the model. This highlights the need for additional structural context or specialized approaches to improve performance on these subgroups. Structural dependency plays a significant role in model performance. For instance, modifications that depend on accessibility or specific structural motifs (e.g., glycosylation) exhibit lower recall.
>
> These findings and analyses will be added to the revised manuscript to clarify the model’s performance across different PTM categories and guide future work to address its limitations. Thank you again for this valuable suggestion!

---

> ### Author Response · Authors · 2024-11-17
> **Response to Reviewer nhjP (4/5)**
>
> **Q7** How are the no-modification sites determined? Are these all sites in protein sequences where no modifications are reported or are they subsets based on amino acid types?
>
> **A7**  The no-modification sites are determined based on residues for which no modifications have been reported in the dbPTM [1] database. This database contains experimentally verified data, with the majority of the reported modification sites being confirmed through MS-based proteomics techniques. Importantly, the no-modification sites in our dataset correspond specifically to those amino acid positions in protein sequences where no modification is listed, rather than being based on particular amino acid types. Therefore, these sites represent regions where modifications have not been experimentally observed in the database, providing a reliable set of non-modified residues for model training and evaluation.
>
> > [1] Li, Zhongyan, Shangfu Li, Mengqi Luo, Jhih-Hua Jhong, Wenshuo Li, Lantian Yao, Yuxuan Pang et al. "dbPTM in 2022: an updated database for exploring regulatory networks and functional associations of protein post-translational modifications." Nucleic acids research 50, no. D1 (2022): D471-D479.
>
> **Q8** Is the quantization technique and the codebook strategy different than the cited work FoldToken? The differences and similarities should be included in the relevant literature part as it is very closely related to the work introduced here.
>
> **A8** Thank you for this insightful question. We appreciate the opportunity to clarify the differences and similarities between our approach and FoldToken.
>
> The main contribution of FoldToken [2] is its SoftCVQ (Soft Conditional Vector Quantization), which aligns the embedding induced by the binary code index with a smooth gradient using an adjustable temperature parameter. Our approach shares this similarity in that we also utilize an adjustable temperature to ensure smooth gradient optimization during training. However, there are key differences in the context and application of these techniques.
>
> While FoldToken is designed for protein backbone generation and requires a highly expressive latent space to effectively reconstruct detailed 3D structures, our work focuses on protein PTM prediction, aiming to learn a compact latent space that captures relevant sequence-structure relationships specific to PTM sites. To achieve this, we introduce a uniform sub-codebook strategy, which helps to more effectively model PTM types by partitioning the latent space into smaller, specialized sub-codebooks for each PTM subtype. This is particularly useful in the context of PTM prediction, where we aim to capture the distinct modification patterns and functional relationships between amino acids at modification sites.
>
> Furthermore, while FoldToken uses a large-scale binary codebook (codebook size = 65536) to allow for detailed protein structure reconstruction, we focus on designing a smaller (codebook size = 26$\times$128=3328), task-specific latent space tailored to PTM representation learning. The sub-codebook strategy in our model ensures that the embeddings focus on the most relevant features for PTM prediction, improving model efficiency and interpretability. Additionally, merely relying on smooth gradient optimization through adjustable temperature is not sufficient for effective PTM prediction, which is why we also incorporate the sub-codebook approach to explicitly guide the model toward the most important features for PTM classification.
>
> We will update the relevant literature section to clearly discuss these similarities and differences, situating our approach within the context of FoldToken and other relevant works in the field, and highlighting how our use of sub-codebooks and compact latent spaces offers a distinct advantage for PTM prediction tasks. We hope this clarifies the differences between our method and FoldToken and we look forward to making these updates in the **Appendix K of our manuscript**. Thank you again for your valuable suggestion.
>
> > [2] Gao, Zhangyang, Cheng Tan, Jue Wang, Yufei Huang, Lirong Wu, and Stan Z. Li. "Foldtoken: Learning protein language via vector quantization and beyond." arXiv preprint arXiv:2403.09673 (2024).

---

> > ### Author Response · Authors · 2024-11-17
> > **Response to Reviewer nhjP (5/5)**
> >
> > **Q9** Of the many structures available for a protein in the PDB, which one is selected? What criteria are applied on this selection?
> >
> > **A9** As illustrated in Appendix C, we prioritize selecting experimentally determined structures from the PDB whenever available, as these are considered the gold standard for structural data. If multiple experimental structures are present for the same protein, we select the one with the highest resolution or the most complete coverage of the protein sequence. In the absence of experimentally determined structures, we turn to high-confidence structural predictions generated by AlphaFold, which have shown to be highly reliable.
> >
> > **Q10** Authors suggest that they introduce a sub-codebook strategy to handle the long-tail distribution but at the end of the day, all the rare classes are grouped as a single class. How does that help?
> >
> > **A10** Thank you for pointing out this concern. We understand that grouping all rare PTM classes into a single "rare sites" category may appear counterintuitive, especially in the context of handling long-tail distributions. However, we argue that this strategy serves several important purposes.
> >
> > The long-tail nature of PTM data is a significant challenge, particularly because rare PTM types have very limited data, with many of these classes having fewer than 20 instances. **All of the rare classes comprise only about 1,000 sites out of the total 1,263,935 sites in the dataset, which represents a very small fraction of the total data (approximately 0.08%)**. Such a disproportionate distribution makes it difficult for models to learn meaningful, class-specific representations for these rare PTMs. By aggregating the rare PTMs into a single class, we effectively increase the representation of this category. This helps the model learn more generalized features and improves its ability to recognize patterns within this otherwise sparse data.
> >
> > Grouping rare PTMs is not purely a form of data aggregation. It is driven by the observation that many rare PTMs, despite differing in their biochemical and biological roles, may share underlying structural or functional properties. For instance, several rare PTMs might be recognized by similar enzymatic machinery or occur in similar structural contexts (e.g., requiring specific residue accessibility or structural motifs). As illustrated in **Figure 6 of our manuscript**, we show that rare PTMs tend to cluster together in the latent space of the model, indicating that they share underlying structural and functional similarities that justify their aggregation. This clustering suggests that the model can capture useful generalizations across rare PTMs, even though they differ in their precise modification mechanisms.
> >
> > We recognize that this approach can affect the evaluation metrics, especially for rare classes. Aggregating rare PTMs into a single class can cause extreme skewing in the metrics, particularly in F1-score, Accuracy, Precision, and Recall, which may become dominated by the more frequent classes. In the case of rare PTMs, such imbalances can lead to anomalous values that are not fully representative of the model's performance across all classes.
> >
> > To address this concern, we conducted additional experiments where rare PTM classes were not merged and were instead treated as distinct categories. This increased the total number of PTM classes from 26 to 73, introducing substantial class imbalance. The results of this experiment are summarized below:
> >
> > | Method | F1     |  Accuracy | Precision | Recall | MCC    | AUROC  | AUPRC  |
> > | -------------------------------- | -------- | --------- | ------ | ------ | ------ | ------ | ------ |
> > | MeToken | 0.5040 | 0.8991   | 0.5808    | 0.4793 | 0.7603 | 0.9220 | 0.5126 |
> > | MeToken w/ all rare PTMs | 0.3045 | 0.9015   | 0.3657    | 0.3031 |  0.7666 | 0.5745 | 0.2335 |
> >
> > ___
> >
> > We greatly appreciate your consideration of these clarifications. Your positive comments in the initial review were very encouraging, and we are truly grateful for your thoughtful feedback. We would be extremely appreciative if you could consider further raising the score or confidence.
> >
> > If you still need any clarification or have any other concerns, please do not hesitate to reach out to us for any reason. Your guidance is greatly appreciated, and we are fully committed to a constructive and productive discussion.

---

> ### Author Response · Authors · 2024-11-21
>
> Dear Reviewer nhjP,
>
> We sincerely appreciate and highly regard your feedback, and we thank your kindness here again. Your contribution works a lot in refining our work.
>
> As the author-reviewer discussion period passes by, we are eager to hear your feedback. Here, we kindly ask if you would consider increasing the score or confidence of our work if we successfully address your concerns. We would like to express our gratitude in advance.
>
> Thanks again here, and your help assists us in improving our work.
>
> Sincerely,
>
> The Authors

---

### Meta-Review · Area_Chair_nP7L · 2024-12-17

**Metareview:**

This paper proposes MeToken, a model that uses a micro-environment token and uniform sub-codebooks to incorporate sequence and structural info for PTM prediction. The reviewers generally liked the motivation and noted some performance gains over existing methods.

The authors did respond to reviewer concerns (extensively!), adding one-hot baselines, clarifying the codebook strategies, discussing structural data advantages, and acknowledging suggestions to improve the paper.

The reviewers did not fully converge on a strongly positive stance. Some appreciated the novelty and thorough experiments, while others felt the paper’s core contribution was not strongly isolated (structural data vs. codebook), or that more analysis was needed. No unanimous positive agreement emerged.

On my personal reading, I am leaning towards acceptance. Some things to consider then: while structural data incorporation is promising, it is not deeply analyzed why it works better than sequence-only. Another common point: the handling of the long-tail distribution and the “rare classes” aggregation raised questions from at least a couple of reviewers about clarity and whether it genuinely solves the problem.

**Additional Comments On Reviewer Discussion:**

see abobe

---

### Decision · Program_Chairs · 2025-01-22

Accept (Poster)